# A Critical Review of the Pharmacokinetics, Pharmacodynamics, and Safety Data of Antibiotics in Avian Species

**DOI:** 10.3390/antibiotics11060741

**Published:** 2022-05-31

**Authors:** Hui Yun Soh, Prisca Xin Yi Tan, Tao Tao Magdeline Ng, Hui Ting Chng, Shangzhe Xie

**Affiliations:** 1Department of Pharmacy, National University of Singapore, Singapore 117559, Singapore; sohhuiyun@protonmail.com (H.Y.S.); priscakyu98@gmail.com (P.X.Y.T.); 2National University of Singapore Libraries, National University of Singapore, Singapore 119275, Singapore; magdeline.ng@nus.edu.sg; 3Mandai Wildlife Group, Conservation, Research, and Veterinary Department, Singapore 259569, Singapore

**Keywords:** antibiotic, bird, dosing

## Abstract

In avian medicine, the use of antibiotic dosing regimens based on species-specific pharmacological studies is ideal. However, due to a lack of such studies, dose extrapolation, which may cause inefficacy and toxicity, is common practice. Multiple searches were performed using the PubMed and Web of Science databases to extract relevant pharmacological studies performed in exotic avian species. The pharmacokinetics (PK), pharmacodynamics (PD), and safety data of the selected antibiotics (enrofloxacin, marbofloxacin, gentamicin, amikacin, ceftiofur, doxycycline, and amoxicillin/clavulanate) from these studies were reviewed. This review aimed to identify trends amenable for safe inter-species dose extrapolation and provide updated findings on dosing regimens that are safe and efficacious for various exotic avian species. We observed that the half-life of antibiotics appears to be shorter in the common ostrich and that amikacin may be amenable to inter-species dose extrapolation as it is safe and shows little inter-species PK and PD variation. Species-specific enrofloxacin dosing regimens that were not listed in the Exotic Animal Formulary (5th ed.) were found for Caribbean flamingos, African penguins, southern crested caracaras, common ostriches, and greater rheas. Specific dosing regimens recommended for psittacine birds (doxycycline 130 mg/kg medicated water) and ratites (PO doxycycline 2–3.5 mg/kg q12 h, PO enrofloxacin 1.5–2.5 mg/kg q12 h and IM enrofloxacin 5 mg/kg q12 h) in the formulary may not be effective in budgerigars and common ostriches, respectively. Apart from the lack of species-specific pharmacological studies, a lack of multiple dose studies was also noted.

## 1. Introduction

Bacterial infections are commonly diagnosed and treated with antibiotics in the avian species. Infected birds can spread zoonotic diseases such as influenza, psittacosis, pasteurellosis, and campylobacteriosis to humans, especially if they are in close contact with them, such as pet-owner settings, zoos, or pet shops [1,2]. The elimination of bacterial infections in birds is therefore important for both human and avian health [3]. For most bacterial infections, selected antibiotics have been recommended for empiric therapy. However, culture sensitivity testing should still be conducted because some bacteria species (such as *Pseudomonas* spp.) are resistant to many antibiotics. In addition, the minimum inhibitory concentrations (MICs) of antibiotics against bacteria from many bird species have not been well studied, and only limited information from a few infected bird species is available [4,5]. This should be investigated as inter-species MIC variability exists and MICs are required to calculate antibiotic efficacy parameters such as C_max_:MIC and T > MIC. In the local setting, antibiograms should be created for exotic birds to help guide antibiotic drug and dosing regimen selection. Currently, most MIC studies have been conducted on bacteria isolated from either the dead bodies of infected and sick birds or from healthy birds [6,7,8,9]. There is a lack of data on the MICs of bacteria isolated from sick but living birds. Moreover, the use of clinical breakpoints is the new recommended standard for guiding antibiotic use. Clinical breakpoints should be based on an epidemiological cut-off value (ECOFF), a PK/PD breakpoint obtained from pre-clinical pharmacokinetic and wherever possible, a clinical cut-off. However, a paucity of data in veterinary medicine, especially avian veterinary medicine, precludes the widespread use of this modality [10].

Currently, some of the antibiotic dosage regimens recommended for use in specific exotic avian species are based on pharmacokinetic (PK) studies conducted in that species. However, the majority are empirically derived or extrapolated from closely related bird species, poultry, or even mammals [11,12,13]. As inter-species variability in anatomy and physiology exists and can be profound even between closely related avian species, drug dosage extrapolation may result in suboptimal drug doses. Subtherapeutic doses result in treatment failure and facilitate the development of resistant strains of bacteria whereas supratherapeutic doses may lead to toxicity and even death. Therefore, it is ideal for dosage regimens to be based on same-species PK studies [11,12,13,14]. At present, the literature evidence on the PK and pharmacodynamic (PD) of antibiotics in different exotic avian species exists as standalone research articles. Therefore, this critical review aimed to (1) collate the PK, PD, and safety data of studies; (2) identify any PK and PD trends in the data that may be amenable to extrapolation across different exotic avian species; and (3) provide recommendations for future practice and research based on the gaps and trends identified.

## 2. Methods

To identify relevant studies, an approach similar to systematic review methodology was undertaken wherein searches were performed on PubMed and Web of Science databases using keywords related to the concept of birds combined with keywords related to the concepts of antibiotics, PK/PD, and adverse effects. The keywords were generated by examining the bird names in the IOC World Bird List Version 10.2 [15], the terminology used in the literature, and the medical subject headings in PubMed. Reference lists from all selected full-text articles were also screened for relevant articles. For a study to be included, it had to provide either PK/PD and/or adverse effects on the antibiotic regimen used. It also had to be on non-poultry avian species, which is defined as any avian species that is not a common poultry species. However, some poultry species commonly kept in zoological collections or outside poultry farms are included (Table 1). The following data were extracted from the studies if available: name of species, sample size, weight, dosing regimen, mean plasma drug concentration, peak plasma drug concentration (C_max_), time to peak plasma drug concentration (T_max_), area under the plasma concentration versus time curve (AUC), mean residence time (MRT), half-life (either elimination or terminal), bioavailability, apparent volume of distribution (V) (if different variants of V was reported, the Vss was extracted), clearance, MIC, the duration plasma drug concentration exceeds MIC (T > MIC), C_max_:MIC, AUC:MIC, adverse effects, and dosing recommendations.

## 3. Drug Disposition and Pharmacokinetic Parameters in Birds

To understand the pharmacokinetic profiles of drugs in avian species, it is important to first understand the pertinent physiological and behavioural traits of birds that can affect the absorption, distribution, metabolism, and excretion of a drug.

The avian stomach is made up of two sections, namely the proventriculus and the gizzard. The proventriculus secretes digestive enzymes while the gizzard is responsible for mechanical digestion and absorption by grinding and crushing food. Birds with softer diets such as carnivorous birds tend to have thin and poorly muscled gizzard walls while insectivorous, herbivorous, and granivorous birds have highly differentiated stomachs with more powerful gizzards, for triturating tougher stomach contents [11]. This could potentially affect drug absorption as efficient mechanical digestion can increase the surface area of the drug presented to the intestines for absorption.

In avian species, the impact of fed or fasted status on the absorption of a drug is accentuated due to the presence of the crop along the oesophagus. In a fasted bird, ingested food bypasses the crop and travels directly into the stomach. However, if the gizzard is filled, food consumed will then be diverted into the crop where there is minimal absorption and stored until contractions discharge food boluses into the lower oesophagus [11]. Absorption mainly occurs in the small intestine [16], and a delayed movement of food from the crop will result in delayed absorption. The digestive system of birds who pursue active prey (raptors) also tend to have shorter intestinal transit times as a reduced ingesta mass aids flight performance [17]. As the transit time determines the duration of the drug at the absorption site, a shorter transit time may lead to a poorer absorption of drugs in birds. The eventual fraction of drug that is systemically absorbed is indicated by the PK parameter of bioavailability, F.

The distribution of a drug largely depends on its physicochemical properties, such as its partition coefficient and acid dissociation constant (pKa). Other factors affecting the distribution of a drug also include the presence of influx or efflux transporters [18], and organ or tissue blood flow [19]. The primary PK parameter, apparent volume of distribution (V), is the fluid volume that a drug seems to be distributed in to account for the observed plasma concentration. It is dependent on the extent of tissue and plasma protein binding [19]. Plasma protein albumin has a high affinity for acidic and neutral drugs, while α1-acid glycoprotein is an important binding protein for basic drugs [20].

The clearance of a drug can be via the metabolism of excretion. In drug metabolism, Phase I reactions add or expose -OH, -SH, -NH_2_, or -COOH functional groups. Phase II reactions couple the drug (or its metabolites) with large, often water-soluble polar molecules [21], so as to render them soluble enough to be excreted in urine. Avian liver enzymes are capable of both Phase I and II reactions [21], which make drugs and their metabolites more water-soluble to be excreted via urine. Relative to mammals, birds tend to have relatively lower levels of cytochrome P450 (CYP450) proteins [22]. This phenomenon is especially witnessed in the marine-eating and raptor species of birds.

In birds, kidney nephrons are mostly reptilian-type, which lack a loop of Henle and are less able to concentrate urine than mammalian-type nephrons [16]. However, avian kidneys are able to vary their glomerular filtration rate (GFR) to a much larger extent than mammals, by up to 65% in states of dehydration [23]. Additionally, birds excrete nitrogenous waste as uric acid [24,25], unlike urea in mammals and humans. This is significant as the excretion of insoluble uric acid reduces water loss and the water requirement of the animal, especially in water-scarce areas. However, in states of renal dysfunction, insoluble uric acid build-up may result in a more rapid manifestation of clinical symptoms such as kidney failure and gout. Avian species also have a renal portal system which is absent in humans, where blood from the caudal half of the body passes through the kidneys before re-joining the systemic circulation [26]. As a result, renally cleared drugs which are intramuscularly administered into the leg may be more rapidly eliminated and result in low bioavailability due to the first-pass effect [27] if it reaches the renal system before entering circulation. This is likely significant as intramuscular injections are usually administered to birds into the pectoral muscle [28]. However, this factor has not been extensively investigated to date.

Clearance (CL) and the apparent volume of distribution (V) of a drug are two primary PK parameters that impact the half-life of a drug. The overall exposure of the animal to a drug can be gleaned from the area under the curve (AUC) parameter.

## 4. Types of Antibiotics

Antibiotics can be classified based on their PD properties [29]. The first group consists of concentration-dependent antibiotics such as the aminoglycosides and fluoroquinolones. Concentration-dependent antibiotics should be given once daily at the highest possible dose as their bactericidal effect increases with increasing C_max_, and they possess a significant post-antibiotic effect (PAE). To achieve clinical response, a C_max_:MIC value of 8–10 is desirable for aminoglycosides [14,30] and an AUC:MIC value of >100–125 h [30] or ≥125 h [14] is desirable for fluoroquinolones according to animal and human clinical studies. The second group consists of time-dependent antibiotics such as beta-lactams. These antibiotics have minimal PAE and their bactericidal action depends on the T > MIC. To achieve an optimal bactericidal effect, T > MIC should be ≥50–80% of the dosing interval [14], or ≥40–50% of the dosing interval [30] according to animal and human studies. The third group consists of antibiotics such as the tetraycyclines and clindamycin. These antibiotics can display time- or concentration-dependent activity depending on the dose used and the pathogen it is targeting. They have a long-to-moderate PAE and their efficacy depends on the AUC:MIC achieved [14,30]. However, the optimal AUC:MIC target is unknown and animal studies have used targets ranging from 13 to 40 h [30,31].

## 5. Included Studies

In this review, we collated data related to fluoroquinolones, aminoglycosies, cephalosporins, and tetracyclines. Details of the included studies such as the number and health status of animals are presented in Appendix A. In general, more than 90% of the studies were pharmacokinetic studies using healthy birds. The concentration achieved was used to predict efficacy based on whether the target MIC would have been achieved or not. This was a relevant measure of efficacy at the time of practice and publication of these studies. The MIC values were typically obtained from other studies by the authors and if the antibiotic targets different microorganism, the highest MIC would be chosen as a surrogate on whether efficacy would be achieved with the concentrations observed. While various PK parameters were extracted and are presented in tables in the following sections, due to the heterogeneity of methods employed in deriving these parameters and the presentation of results by the various studies, the discussion for PK trends will centre around the half-life of drugs, since it is a composite of CL and V.

## 6. Fluoroquinolones

Fluoroquinolones are broad-spectrum antibiotics which have potent bactericidal activity against Gram-negative *Enterobacteriaceae*. Some also have activity against selected Gram-positive bacteria and *Pseudomonas* spp. In humans, most fluoroquinolones are well-absorbed orally, have high volumes of distribution, and are predominantly cleared by the kidneys [32]. In avian medicine, enrofloxacin and marbofloxacin are the two most commonly used [12,33] and studied fluoroquinolones. They are used to treat avian respiratory and alimentary diseases caused by bacteria such as *Chlamydia psittaci*, *Pseudomonas aeruginosa*, and *Salmonella* spp.

### 6.1. Enrofloxacin

The adverse effects and dosing recommendations of enrofloxacin are presented in Table 2 and the PK data are presented in Table 3. Enrofloxacin appears to be safe for exotic birds as most studies did not report any adverse effect and those that did only reported mild reversible adverse effects.

The half-life of enrofloxacin varied across the different species from as short as 0.783 ± 0.06 h to as long as 19.4 h. For the same IM 15 mg/kg single dose, the enrofloxacin half-life is similar between the greater rhea (2.85 ± 0.54 h) [34], common ostrich (2.36 ± 0.66 h) [35], and African grey parrot (2.31 h) [36]. The similarity in enrofloxacin half-life between the common ostrich and greater rhea may explained by the fact that they are closely related species (both species are ratites). For the same IM 15 mg/kg single dose, the enrofloxacin half-life is similar between the great horned owl (11.4 h) [37] and red-tailed hawk (11.0 h) [37], even though they are not closely related species. The similarity in their drug half-lives cannot be explained by the similarity in their weight (0.96–1.33 kg for the great horned owl and 0.96–1.54 kg for the red-tailed hawk) because the weights of the greater rhea, common ostrich, and African grey parrot differ widely (0.47–99 kg); however, they have similar enrofloxacin half-lives (2.31 to 2.85 h). This highlights the need for exercising caution when extrapolating IM enrofloxacin doses between different bird species based on weight. However, there is a possibility that IM enrofloxacin doses may be extrapolated between closely related bird species. Studies should be conducted on more ratite species to verify this.

Generally, the half-life seems to be shorter in the ratite species than in the other bird species regardless of the dosing regimen used. Young ostriches have shorter enrofloxacin half-lives than adult ostriches, possibly due to age-related differences in drug elimination.

In one study, African grey parrots given an oral multiple dose treatment regimen had a significant decrease in mean plasma concentration, AUC, MRT, and half-life at the end of the treatment, and these reductions were substantial enough to justify considering a dose increase to achieve effective plasma concentration for moderately resistant organisms. This change in enrofloxacin disposition was attributed to the increased metabolism of enrofloxacin to ciprofloxacin [36]. This trend of a shortening enrofloxacin half-life over multiple oral doses seems to be unique in the African grey parrot as enrofloxacin has been shown to inhibit CYP450 enzymes in chickens [38], sea bass [39], and crucian carp [40] without causing auto-induction. The metabolism of enrofloxacin has been attributed to CYP3A enzymes in broilers [41] but more studies should be conducted to ascertain the enzyme responsible for the metabolism of enrofloxacin in African grey parrots and whether there is any auto-induction causing the shortening of its half-life. This trend highlights the importance of conducting multiple-dose studies, especially since antibiotics are given in multiple doses during a period of time and the PK of a drug may be significantly different after multiple doses due to factors such as CYP enzyme induction.

Red-tailed hawks and great horned owls appear to have a similar drug exposure, as evidenced by their similar AUCs of 44 and 47.2 μg·h/mL, respectively, after being orally administered a single dose of 15 mg/kg enrofloxacin, even though the former species has a crop while the latter does not [37]. This shows that differences in gastrointestinal anatomy may not always result in an expected difference in drug PK, and thus caution should be exercised when extrapolating doses to unstudied species on the basis of reasoning based upon differences in physiology. That said, T_max_ of 7.1 h in great horned owl and 5.4 h in red-tailed hawk appear to correspond with the trend of raptors with slower GI transit times than other bird species [42]. However, it has been shown in one study that the GI transit time of red-tailed hawks is shorter than that of various psittacine birds [43]. The T_max_ values obtained after drug administration by the PO route for the red-tailed hawk and great horned owl are longer than that obtained for other bird species [42].

For the five bird species given an IM 15 mg/kg dose of enrofloxacin, the AUC values appear to be positively correlated with half-life values, and as expected with longer half-lives, there is greater systemic exposure to the drug [34,35,36,37].

**Table 2 antibiotics-11-00741-t002:** Adverse effects, efficacy parameters, and dosage recommendations of enrofloxacin in exotic avian species.

Dosage	Species	MIC (μg/mL)	C_max_:MIC	AUC:MIC (h)	Adverse Effects	Dosage Recommended by Authors	Ref.
**Intramuscular**							
5 mg/kg SD	Common ostrich (young)	0.03 0.06 0.1 0.13	14.50 ± 4.12 7.25 ± 2.06 4.35 ± 1.24 3.35 ± 0.95	30.42 ± 4.00 15.25 ± 1.24 9.13 ± 1.20 7.02 ± 0.92	Not reported	All AUC:MIC achieved was lower than the target of 125:1	[44]
10 mg/kg SD	Houbara bustard	0.5 2.0	5.44 1.36	55 13.7	Elevated CK, LDH and AST were observed at 12 h, likely caused by restraint	-	[4]
	Southern crested caracara	0.25	15.94	139.63	None observed	IM 9.5 mg/kg q24 h for MIC = 0.25 μg/mL	[45]
15 mg/kg SD	African grey parrot	-	-	-	Not reported	-	[36]
	Great horned owl	1	3.8 ^b^	65.3 ^b^	None observed	IM 15 mg/kg q24 h	[37]
	Greater rhea	0.1 0.03	35.5 ± 9.52 118 ± 31	48.25 ± 7.95 161 ± 26	Not reported	IM 15 mg/kg	[34]
	Red-tailed hawk	1	4.5 ^b^	54 ^b^	None observed	IM 15 mg/kg q24 h	[37]
15 mg/kg q12 h × 3 days	Common ostrich (adult)	0.06 0.25	18.86 ± 7.74 4.53 ± 1.85	257.78 ± 45.34 61.87 ± 34.88	CK activity increased by 2–9-fold compared with pre-administration concentration; this may be due to muscle damage	SC 15 mg/kg q12 h	[35]
**Intravenous**							
2.2 ± 0.03 mg/kg SD	Emu	0.5	82.36 ^b^	16.52 ^b^	Not reported	IV 2.2 mg/kg q12 h	[46]
5 mg/kg SD	Common ostrich	0.03 0.06 0.1 0.13	54.42 ± 1.66 27.21 ± 0.83 16.32 ± 0.50 12.56 ± 0.38	32.04 ± 3.34 16.02 ± 1.67 9.61 ± 1.0 7.39 ± 0.77	Not reported	All AUC:MIC achieved was lower than the target of 125:1	[44]
	Southern crested caracara	0.25	13.74	90.73	None observed	IV 7.5 mg/kg q24 h for MIC = 0.25 μg/mL	[45]
7.5 mg/kg SD	Great horned owl	1	-	-	2 owls immediately developed bradycardia, peripheral vasoconstriction, and stupor. The timing and nature of response was suggestive of anaphylaxis ^a^	IV administration of enrofloxacin should not be attempted in great horned owls	[37]
10 mg/kg SD	Houbara bustard	0.5 2.0	9.78 2.44	59.6 14.9	None observed	10 mg/kg given PO or parenterally q12 h, OR 15 mg/kg given parenterally q24 h	[4]
15 mg/kg SD	African penguin	0.5	15.72 ^b^	165 ^b^	Not reported	-	[42]
	Greater rhea	-	-	-	Not reported	-	[47]
	Red-tailed hawk	1	6.70 ^b^	62.2 ^b^	None observed	Use with caution	[37]
**Subcutaneous**							
15 mg/kg SD	Caribbean flamingo	0.25 0.5	≥8 ≥8 (but only 3/7 birds > 4 μg/mL)	>100 <100 (but 3 of 7 birds had AUC > 50 μg·h/mL)	None observed	15 mg/kg PO or SC Q24 h suitable for bacteria with MIC ≤ 0.25 μg/mL	[48]
15 mg/kg q12 h × 3 days	Common ostrich (adult)	0.06 0.25	13.07 ± 2.63 3.13 ± 0.63	210.7 ± 52.98 50.44 ± 12.71	None observed	SC 15 mg/kg q12 h	[35]
**Oral**							
3 mg/kg SD crop gavage	African grey parrot	-	-	-	Not reported	-	[36]
10 mg/kg SD	Houbara bustard	0.5 2.0	3.68 0.92	36.6 9.15	Elevated CK, LDH, and AST activity were observed at 12 h, likely caused by restraint	-	[4]
15 mg/kg SD crop gavage	African grey parrot	-	-	-	Not reported	15 mg/kg PO bd	[36]
15 mg/kg SD pill in fish	African penguin	0.5	>8 for 5/6 birds, last bird ratio of 7.4	>125 in 4/6 birds, other 2 had ratio of 115 and 108	Well tolerated (no adverse effects reported)	15 mg/kg PO in fish or pill q24 h	[42]
15 mg/kg SD pill	African penguin	0.5	>8 for 4/6 birds, other 2 had ratio of 5.9 and 7.6	>125 in 6/6 birds	Well tolerated (no adverse effects reported)	15 mg/kg PO in fish or pill q24 h	
15 mg/kg SD oesophagus gavage	Caribbean flamingo	0.25 0.5	≥8 ≥8	>100 <100 (but 5/7 birds had above 50 μg·h/mL)	None observed	15 mg/kg SC or PO recommended for bacteria with MIC ≤ 0.25 μg/mL	[48]
15 mg/kg SD in prey	Great horned owl	1	2.6 ^b^	44 ^b^	None observed	PO 15 mg/kg in prey q24 h	[37]
	Red-tailed hawk	1	2.8 ^b^	47.2 ^b^	None observed		
30 mg/kg SD crop gavage	African grey parrot	-	-	-	Not reported	-	[36]
30 mg/kg q12 h × 10 days crop gavage	African grey parrot	-	-	-	Water consumption increased; 3 birds became slightly polyuric and 1 bird became markedly polyuric; polyuria resolved 2–3 days after treatment ended	-	
0.09, 0.19, 0.38, 0.75, 1.5 and 3.0 mg/mL × 7 days medicated water	African grey parrot	-	Only trough concentration measured	-	Acceptance of the water at doses of 1.5 and 3.0 mg/mL was unsatisfactory	-	[49]

**SD** = single dose; **MIC** = minimum inhibitory concentration (please refer to the original article for details on the specific pathogen being targeted); **C_max_:MIC** = ratio of peak plasma drug concentration to minimum inhibitory concentration; **AUC:MIC** = ratio of area under the plasma concentration versus time curve to minimum inhibitory concentration. ^**a**^: In mammals, an anaphylaxis reaction usually includes hypotension and vasodilation. In this case, hypotension and vasoconstriction occurred. The inflammatory cascade in birds possibly includes vasoconstriction instead of vasodilation. Alternatively, the response could have been a non-immunologic one, occurring as a result of the pharmacodynamics effects of enrofloxacin or its carrier. ^**b**^: calculated using the MIC, C_max_, and AUC data provided in study.

**Table 3 antibiotics-11-00741-t003:** Pharmacokinetic values for enrofloxacin in exotic avian species.

Dosage	Species	Weight (kg)	C_max_ (μg/mL)	T_max_ (h)	Half-Life (h)	AUC (μg·h/mL)	V (L/kg)	Clearance (mL/min/kg)	BA (%)	Ref.
**Intramuscular**										
5 mg/kg SD	Common ostrich (young)	34–53	0.44 ± 0.12	1.05 ± 0.57	0.867 ± 0.367	0.91 ± 0.12	-	-	91 ± 5	[44]
10 mg/kg SD	Houbara bustard	0.945–1.655	2.75 ± 0.11	1.72 ± 0.19	6.39 ± 1.49	27.5 ± 3.67	3.18 ± 0.32	6.56 ± 0.95	97.3	[4]
	Southern crested caracara	1.33 ± 0.06	3.92	0.72	6.58	21.92	-	-	78.76	[45]
15 mg/kg SD	African grey parrot	0.47–0.55	3.87	1	2.31	13.28	-	0.0188 ^a^	-	[36]
	Great horned owl	0.96–1.33	3.8	2.1	11.4	65.3	3.4	-	-	[37]
	Greater rhea	3.35 ± 0.34	3.30 ± 0.90	0.403 ± 0.153	2.85 ± 0.54	4.18 ± 0.69	-	-	-	[34]
	Red-tailed hawk	0.96–1.54	4.5	1.1	11.0	54.0	2.4	-	87	[37]
	Common ostrich (adult)	84–99	1.09 ± 0.38	2.41 ± 1.2	2.36 ± 0.66	6.33 ± 2.15	-	-	-	[35]
15 mg/kg q12 h × 3 days	Common ostrich (adult)	84–99	1.13 ± 0.47	-	-	-	-	-	-	
**Intravenous**										
2.2 ± 0.03 mg/kg SD	Emu	25–26	3.25, 15.09 and 104.40 for the 3 emus	0	3.33	8.26	0.0584 ^c^	6.00 ^b^	-	[46]
5 mg/kg SD	Common ostrich (young)	34–53	1.6 ± 0.05	0	0.783 ± 0.06	0.96 ± 0.10	3.4 ± 0.41	76 ± 5.3	-	[44]
	Southern crested caracara	1.33 ± 0.06	-	-	7.81	34.38	2.3	-	-	[45]
10 mg/kg SD	Houbara bustard	0.945–1.638	-	-	5.63 ± 0.54	29.8 ± 1.74	2.98 ± 0.32	5.71 ± 0.41	-	[4]
15 mg/kg SD	African penguin	3.6 ± 0.57	7.86	0	13.7	82.5	3	3.03	-	[42]
	Greater rhea	2.97 ± 0.26	0.27 ± 0.07	0.37 ± 0.14	2.66	3.57 ^d^	5.01	65.8 ^b^	-	[47]
	Red-tailed hawk	0.96–1.54	6.7	-	19.4	62.2	2.3	-	-	[37]
**Subcutaneous**										
15 mg/kg SD	Caribbean flamingo	2.2–3.6	5.77	1.1	6.46	47.3		5.28		[48]
	Common ostrich (adult)	84–99	1.51 ± 0.79	1.45 ± 0.6	3.19 ± 1.09	8.15 ± 2.05	-	-	-	[35]
15 mg/kg q12 h × 3 days	Common ostrich (adult)	84–99	0.78 ± 0.16	-	-	-	-	-	-	
**Oral**										
3 mg/kg SD crop gavage	African grey parrot	0.47–0.55	0.32	2–4	2.59	1.80	-	0.0323 ^a^	-	[36]
10 mg/kg SD	Houbara bustard	0.945–1.655	1.84 ± 0.16	0.66 ± 0.05	6.80 ± 0.79	18.3 ± 1.81	5.12 ± 0.51	9.21 ± 0.94	62.7 ± 11.1	[4]
15 mg/kg SD crop gavage	African grey parrot	0.47–0.55	1.12	2–4	2.52	6.73	-	0.0372 ^a^	-	[36]
15 mg/kg SD pill in fish	African penguin	3.6 ± 0.57	4.77	1.59	11.9	80.5	-	3.05	-	[42]
15 mg/kg SD pill	African penguin	3.6 ± 0.57	4.38	4.80	13.8	92.9	-	2.67	-	
15 mg/kg SD oesophagus gavage	Caribbean flamingo	2.2–3.6	5.25	1.49	5.83	49.9	-	5.01	-	[48]
15 mg/kg SD in prey	Great horned owl	0.96–1.33	2.6	7.1	7.2	44.0	4.2	-	-	[37]
	Red-tailed hawk	0.96–1.54	2.8	5.4	8.9	47.2	4.2		76	
30 mg/kg SD crop gavage	African grey parrot	0.47–0.55	1.69	2–4	2.74	12.84	-	0.039 ^a^	-	[36]
30 mg/kg q12 h × 10 days crop gavage	African grey parrot	0.47–0.55	-	-	2.96 (first interval) 1.90 (last interval)	1.90	-	-	-	
0.09, 0.19, 0.38, 0.75, 1.5 and 3.0 mg/mL × 7 days medicated water	African grey parrot	0.470–0.550	-	-	-	-	-	-	-	[49]

**SD** = single dose; **BA** = bioavailability; **C_max_** = peak plasma drug concentration; **T_max_** = time to peak plasma drug concentration; **AUC** = area under the plasma concentration versus time curve; **V** = apparent volume of distribution. ^**a**^: units converted from mL/kg/h to mL/kg/min by dividing the original value reported by 60. ^**b**^: units converted from L/h/kg to mL/kg/min by multiplying the original value by 1000 and then dividing it by 60. ^**c**^: units converted from L to L/kg by dividing the original value by the average weight of the bird. ^**d**^: units converted from mg·h·L to μg·h/mL by dividing the original value by 1000 and then multiplying by 1000.

### 6.2. Marbofloxacin

The adverse effects and dosing recommendations of marbofloxacin are presented in Table 4 and the PK data are presented in Table 5.

Marbofloxacin seems to be safe for exotic birds as no adverse effect was reported in any study.

The half-life of marbofloxacin in the Eurasian buzzard [50] was 3-fold longer than that in the Eurasian griffon vulture (EGV) [51] even though both species were given the same dosing regimen and belong to the Accipitriformes order. This difference could be due to the EGV having a low drug clearance rate due to its lower heart rate and metabolic rate. The half-life of the Eurasian buzzard is closer to that of the blue-and-gold macaw [52], even though the latter belongs to the Psittaciformes order. Thus, caution has to be taken even when extrapolating drug doses between closely related avian species. Marbofloxacin half-life is shortest in the common ostrich [53], among the avian species studied.

**Table 4 antibiotics-11-00741-t004:** Adverse effects, efficacy parameters, and dosage recommendations of marbofloxacin in exotic avian species.

Dosage	Species	MIC (μg/mL)	C_max_:MIC	AUC:MIC (h)	Adverse Effects	Dosage Recommended by Authors	Ref.
**Intramuscular**							
5 mg/kg SD	Common ostrich	0.2 0.02	5.63 56.33	11.26 112.60	Not reported	-	[53]
**Intravenous**							
2 mg/kg SD	Eurasian buzzards	-	-	-	None observed	IV 2 mg/kg q12 h	[54]
	Eurasian griffon vulture	0.2	-	97.3 ^a^	Not reported	IV 2.73 mg/kg q24 h	[51]
2.5 mg/kg SD	Blue-and-gold macaw	1	-	9.41 ± 2.84 ^a^	None observed	PO 2.5 mg/kg q24 h	[52]
5 mg/kg SD	Common ostrich	0.2 0.02	17.07 170.67	11.58 115.84	Not reported	-	[53]
	Common pheasant	-	-	-	Not reported	-	[55]
**Intraosseous**							
2 mg/kg SD	Eurasian buzzard	0.1	19.2	85.3	None observed	-	[50]
**Oral**							
crop gavage 2.5 mg/kg SD	Blue-and-gold macaw	1	1.08 ± 0.316 ^a^	7.94 ± 2.08 ^a^	None observed	PO 2.5 mg/kg q24 h	[52]
	Common pheasant	-	-	-	Not reported	-	[55]
crop gavage 10 mg/kg SD	Eurasian buzzard	0.25	14.8	181.75	None observed	PO 10 mg/kg OD	[56]

**SD** = single dose; **MIC** = minimum inhibitory concentration; **C_max_:MIC** = ratio of peak plasma drug concentration to minimum inhibitory concentration; **AUC:MIC** = ratio of area under the plasma concentration versus time curve to minimum inhibitory concentration. ^**a**^: calculated using MIC, C_max_, AUC data provided in study.

**Table 5 antibiotics-11-00741-t005:** Pharmacokinetic values for marbofloxacin in exotic avian species.

Dosage	Species	Weight (kg)	Half-Life (h)	C_max_ (μg/mL)	T_max_ (h)	V (L/kg)	CL (mL/min/kg)	Ref.
**Intramuscular**								
5 mg/kg SD	Common ostrich	43.48 ± 6.02	1.96 ± 0.35	1.13 ± 0.29	0.61 ± 0.36	-	-	[53]
2 mg/kg SD	Eurasian buzzards	0.730–1	4.11 ± 0.77	-	-	1.16 ± 0.21	3.98 ^b^	[54]
2 mg/kg SD	Eurasian griffon vulture	6–8.5	12.51 ± 2.52	-	-	0.208 ± 0.0303 ^a^	1.82 ± 0.383 ^c^	[51]
2.5 mg/kg SD	Blue-and-gold macaw	1.04	4.3	-	-	1.3 ± 0.32	4.83 ± 1.3 ^c^	[52]
5 mg/kg SD	Common ostrich	43.48 ± 6.02	1.47 ± 0.31			0.0740 ± 0.022 ^a^	48.5 ± 4.5 ^c^	[53]
	Common pheasant	1.0 ± 0.04	8.4 ± 0.74			1.4 ± 0.11	3.17 ± 0.217 ^c^	
**Intraosseous**								
2 mg/kg SD	Eurasian buzzard	0.730–0.875	4.91 ± 0.65	1.92 ± 0.78	0.11 ± 0.08	-	0.0557 ± 0.002 ^d^	[50]
**Oral**								
Crop gavage 2.5 mg/kg SD	Blue-and-gold macaw	1.04	3.9	1.08 ± 0.316	2.6 ± 0.8	-	-	[52]
	Common pheasant	1.0 ± 0.04	6.6 ± 3.10	3.0 ± 0.27	5.7 ± 0.62	-	-	
Crop gavage 10 mg/kg SD	Eurasian buzzard	0.6–0.9	9.48	3.70	2.92	-	-	[56]

**SD** = single dose; **C_max_** = peak plasma drug concentration; **T_max_** = time to peak plasma drug concentration; **V** = apparent volume of distribution; **MRT** = mean residence time. ^**a**^: units converted from L to L/kg by dividing the original value by the average weight of the species. ^**b**^: units converted from mL/min to mL/min/kg by dividing the original value by the average weight of the species. ^**c**^: units converted from L/h/kg to mL/min/kg by multiplying the original value by 1000 and then dividing by 60. ^**d**^: units converted from mL/h/kg to mL/min/kg by dividing the original value by 60.

## 7. Aminoglycosides

Aminoglycosides are bactericidal agents that are primarily active against aerobic Gram-negative bacilli such as *Escherichia coli* and *Pseudomonas aeruginosa*. In humans, aminoglycosides are poorly absorbed from the gastrointestinal (GI) tract as they are polar cations but are well intramuscularly absorbed. These drugs do not undergo significant metabolism, are primarily excreted by renal filtration, and have a low volume of distribution [32]. In avian medicine, amikacin and gentamicin are the two most commonly used and studied aminoglycosides as amikacin produces the least adverse effect and gentamicin is the least costly [12,33].

### 7.1. Gentamicin

The adverse effects and dosing recommendations of gentamicin are presented in Table 6 and the PK data are presented in Table 7. In addition to negligible injection site injury, no other adverse effect was observed in any of the single-dose studies [57]. However, in the multiple dose studies, serious adverse effects such as anorexia, depression, neuromuscular blockade and death were observed in birds of prey (great horned owl, lanner falcon, and red-tailed hawk) [58,59,60]. Nephrotoxicity, which is a known adverse effect of gentamicin, has also been observed in red-tailed hawks and lanner falcons [59,60]. Future studies could investigate the safety and efficacy of lower dose regimens in birds of prey.

The same multiple dose regimen resulted in variable adverse effects in the great horned owl, ranging from death within 3 days to no clinical sign at any point within the trial [58]. This study highlights the impact of intra-species differences in factors such as age, sex, and weight on drug PK, PD, and adverse effects.

The adverse effects that occurred in the multi-dose studies on psittacine birds showed inter-species variation in severity [61,62]. Despite having similar PK parameter values, galahs experienced greater toxicity than scarlet macaws, even though only the macaws had evidence of a change in PK when the results of the single and multiple dose studies were compared. Namely, the plasma drug concentration at 12 h on treatment days 3, 5, and 7 was significantly higher in the multiple dose regimen for the macaws. This suggests species-specific differences in susceptibility to gentamicin-induced toxicity, regardless of drug disposition [62].

**Table 6 antibiotics-11-00741-t006:** Adverse effects and efficacy parameters of gentamicin in exotic avian species.

Dosage	Species	MIC (μg/mL)	C_max_:MIC (μg/mL)	Adverse Effects	Ref.
**Intramuscular**					
5 mg/kg SD	Budgerigar	-	-	Negligible muscle injury at the injection site. Higher doses resulted in increased injury.	[57]
	Galah	-	-	Not reported.	[62]
	Scarlet macaw	-	-	Not reported.	
	Greater sandhill crane	-	-	Not reported. IM 5 mg/kg q8 h will produce potentially toxic plasma concentrations for approximately 1 h following injection.	[63]
	Ring-necked pheasant	-	-	Not reported. IM 5 mg/kg q8 h will produce potentially toxic plasma concentrations for approximately 1 h following injection.	
5 mg/kg q24 h × 4 days	Lanner falcon	-	-	The birds had severe respiratory tract infection. On day 4 of treatment, both birds developed an acute adverse reaction to the drug and treatment was discontinued. Muscular spasm, loss of balance, and apparent impaired vision were observed. The birds were immediately held and massaged and returned to “normal state” after a few minutes. Droppings were collected from the falcons to evaluate renal function and compared to those of a healthy sharp shinned hawk. The falcon droppings had excessively high NABG activity, suggestive of renal toxicosis. However, it is unlikely that the neuromuscular and ototoxic-like signs were related to the treatment. It is more probable these signs were induced by the apparent fear of falcons to the injection of the drug.	[60]
5 mg/kg q12 h × 3 days	Cockatiel	-	-	Weight loss (6.25–18.7%) due to handling and restraint.	[61]
5 mg/kg q12 h × 7 days	Galah	-	-	High plasma AST for at least the first 7 days of treatment but were normal by day 21. Plasma LD was high immediately after treatment but was normal by day 7 of treatment. Polydipsia was observed on day 3 and persisted for 23 d after termination of treatment. Polyuria was observed on day 4 and persisted for 30 days after treatment ended.	[62]
5 mg/kg q12 h × 7 days	Scarlet macaw	-	-	Plasma AST was high at 12 h after treatment, but normal by day 17. Water consumption was difficult to determine as there was much spillage. However, there was no significant difference between the treatment and control group in terms of water consumption for 11 of the first 15 days of the study. Slight polyuria was observed on day 4 of treatment but urine output returned to normal within 5 days after treatment ended.	
10 mg/kg SD	Budgerigar			Negligible muscle injury at the injection site. Higher doses resulted in increased injury.	[57]
	Galah	4	9.36 ^a^	Not reported.	[62]
	Scarlet macaw	4	9.36 ^a^	Not reported.	
	Golden eagle	-	-	Not reported.	[64]
	Great horned owl	-	-	Not reported.	
	Greater sandhill crane	-	-	Not reported.	[63]
	Red-tailed hawk	-	-	Not reported.	[64]
	Ring-necked pheasant	-	-	Not reported.	[63]
10 mg/kg q12 h × 5 days	Great horned owl	-	-	In the treatment group (n = 8), 6 owls developed polyuria, 3 became anorexic and depressed, and 2 eventually died. Clinically normal behaviour in surviving owls required 24 h–4 weeks to become re-established. Response to gentamicin greatly varies, from death within 3 days to no clinical sign at any point within the trial.	[58]
20 mg/kg SD	Greater sandhill crane	-	-	Not reported.	[63]
	Ring-necked pheasant	-	-	Not reported.	
**Intravenous**					
10 mg/kg SD	Golden eagle	-	-	Not reported.	[64]
	Great horned owl	-	-	Not reported.	
	Red-tailed hawk	-	-	Not reported.	
10 mg/kg q12 h × 4 days	Red-tailed hawk	-	-	Clinical signs and water consumption were variable. Sex, age, and renal function can all contribute to intraspecies variation, but no clear correlation was found among any of these factors and response. This dose is considered toxic to red-tailed hawks. The most marked changes were seen on day 4. On day 4, 1 bird had slight depression, 1 was ataxic, 1 was very depressed and dyspnoeic, and 2 were normal; 2 birds and 3 birds had increased and decreased water consumption, respectively. Total protein, BUN, albumin, and uric acid significantly increased and ALP significantly decreased. Two birds were euthanized. The kidneys of one bird contained urate crystals and the liver surface of the other bird had a mottled appearance and focal petechial haemorrhages.	[59]
20 mg/kg q12 h × 6 days	Red-tailed hawk	-	-	Clinical signs and water consumption were variable. Sex, age and renal function can all contribute to intraspecies variation, but no clear correlation was found among any of these factors and response. This dose is considered toxic to red-tailed hawks. Clinical signs that occurred after at least 1 day of drug administration were depression, weakness, dyspnoea, hypothermia, apnoea, and death. Initially, the birds recovered at least partially within 8 h but clinical signs recurred and often became more severe with the next injection. Water consumption decreased in 1 bird but was unchanged in the other 4 birds. On day 4, ALT, cholesterol, total protein, and albumin had significantly increased. AST increased by 4-fold in 3 birds and by 1000-fold in 1 bird. All birds eventually died: 1 bird died each on days 1, 3, and 6 and 2 birds died on day 2. Upon examination of tissues, 1 bird had visceral gout, 2 birds had focal caseous air sacculitis, and 3 birds had urate deposits in the kidneys. Weakness, dyspnoea, and sudden death after injection may be attributed to neuromuscular blockade. It is possible that red-tailed hawks are more susceptible to gentamicin-induced neuromuscular blockade.	

**SD** = single dose; **C_max_** = peak plasma drug concentration; **NABG** = N-acetyl-β-glucosaminidase; **AST** = aspartate transaminase; **LD** = L-lactate dehydrogenase; **ALP** = alkaline phosphatase; **BUN** = blood urea nitrogen; **ALT** = alanine transaminase. ^**a**^: calculated using MIC, C_max_, and AUC data provided in this study. For C_max_ value, the mean peak plasma concentration at 30 min was used.

**Table 7 antibiotics-11-00741-t007:** Pharmacokinetic values of gentamicin in exotic avian species.

Dosage	Species	Weight (kg)	Half-Life (h)	C_max_ (μg/mL)	T_max_ (h)	V (L/kg)	Clearance (mL/kg/min)	Ref.
**Intramuscular**								
5 mg/kg SD	Budgerigar	0.025–0.035 ^a^	0.53	17.3	0.25	-	-	[57]
	Galah	0.310–0.510	1.23	-	-	0.216	0.0337 ^b^	[62]
	Greater sandhill crane	3.6–5.3	2.74 ± 0.617	-	-	-	-	[63]
	Ring-necked pheasant	0.9–1.5	1.25 ± 0.252	-	-	-	-	
	Scarlet macaw	0.750–1.05	1.17	-	-	0.176	0.029 ^b^	[62]
5 mg/kg q12 h × 7 days	Scarlet macaw	0.750–1.05	-	-	-	-	-	
5 mg/kg q24 h × 4 days	Lanner falcon	0.5–0.9 ^a^	-	-	-	-	-	[60]
5 mg/kg q12 h × 7 days	Galah	0.310–0.510	-	-	-	-	-	[62]
5 mg/kg q12 h × 3 days	Cockatiel	0.081–0.136	1.18	4.66 ± 1.45	1.5	-	-	[61]
10 mg/kg SD	Budgerigar	0.025–0.035 ^a^	0.53	37.0	0.25	-	-	[57]
	Galah	0.310–0.510	1.44	37.44 ± 2.60	0.5		0.0332 ^b^	[62]
	Golden eagle	2.73–4.42	2.46 ± 0.32	35	0.5	0.21 ± 0.01	1.01 ± 0.06	[64]
	Great horned owl	1.09–2.02	1.93 ± 0.24	35	0.5	0.23 ± 0.02	1.41 ± 0.10	
	Greater sandhill crane	3.6–5.3	2.74 ± 0.617	37.5 ^c^	1 ^c^	-	-	[63]
	Red-tailed hawk	0.94–1.71	1.35 ± 0.18	35	0.5	0.24 ± 0.03	2.09 ± 0.16	[64]
	Ring-necked pheasant	0.9–1.5	1.25 ± 0.252	35.3 ± 11.4 ^c^	0.75 ^c^	-	-	[63]
	Scarlet macaw	0.750–1.05	1.07	37.44 ± 2.60	0.5	0.172	0.031 ^b^	[62]
10 mg/kg q12 h × 5 days	Great horned owl	0.910–2.5 ^a^	-	-	-	-	-	[58]
20 mg/kg SD	Greater sandhill crane	3.6–5.3	2.74 ± 0.617	-	-	-	-	[63]
	Ring-necked pheasant	0.9–1.5	1.25 ± 0.252	-	-	-	-	
**Intravenous**								
10 mg/kg SD	Golden eagle	2.73–4.42	2.46 ± 0.32	48	0.5	0.21 ± 0.01	1.01 ± 0.06	[64]
	Great horned owl	1.09–2.02	1.93 ± 0.24	39	0.5	0.23 ± 0.02	1.41 ± 0.10	
	Red-tailed hawk	0.94–1.71	1.35 ± 0.18	35	0.5	0.24 ± 0.03	2.09 ± 0.16	
10 mg/kg q12 h × 4 days	Red-tailed hawk	0.690–1.460 ^a^	-	-	-	-	-	[59]
20 mg/kg q12 h × 6 days	Red-tailed hawk	0.690–1.460 ^a^	-	-	-	-	-	

**SD** = single dose; **C_max_** = peak plasma drug concentration; **T_max_** = time to peak plasma drug concentration; **V** = apparent volume of distribution. ^**a**^: weight of great horned owl and red-tailed hawk were obtained from https://www.allaboutbirds.org (accessed on 1 May 2022); weight of budgerigar and lanner falcon were obtained from https://lafeber.com/vet/basic-information-sheet-for-the-parakeet/ (accessed on 1 May 2022) and https://www.torontozoo.com/animals/Lanner%20Falcon (accessed on 1 May 2022), respectively. ^**b**^: units were converted from mL/kg/h to mL/kg/min by dividing the original value by 60. ^**c**^: C_max_ not stated by author of the study. C_max_ is assumed to be the highest reported plasma drug concentration. The plasma drug concentration of ring-necked pheasants was 27.9 ± 11.7, 29.8 ± 12.4, 35.3 ± 11.4, and 23.3 ± 9.1 μg/mL at 15, 30, 45, and 60 min following drug administration, respectively. The plasma drug concentration of the greater sandhill cranes was 29.2, 32.7, and 37.5 μg/mL at 15, 30, and 60 min following drug administration, respectively.

The half-life of a drug is dependent on the two primary PK parameters, namely clearance and volume of distribution. Variability in either of these parameters could account for the differences in the half-life of a drug in different species. In general, it was observed that half-life seems to be correlated with weight, with heavier species having longer half-lives than lighter ones [57,61,62,63,64]. This is consistent with the understanding that larger animals tend to have slower drug elimination than smaller animals due to a decreased relative liver and kidney size in comparison to their body size [65,66]. However, this factor alone does not account for the variabilities observed as there were also data showing that inter-species half-life differences were not always proportional to weight differences. For example, among psittacine species, for the same IM 5 mg/kg dose, the half-life of gentamicin in the galah, scarlet macaw, and cockatiel is very similar (1.17 h–1.23 h) even though they have different weights (0.081–1.05 kg). However, the half-life in the budgerigar (0.025–0.035 kg) is much shorter (0.53 h). Perhaps the differences in half-life could then be due to different fat and water body composition that may affect the distribution of a drug. Within the same species, gentamicin appears to exhibit linear kinetics as the half-life is constant regardless of the dose and route of administration. In humans, aminoglycosides also exhibit linear kinetics.

The C_max_ obtained for cockatiels was much lower (4.66 μg/mL) than that for budgerigars (17.3 μg/mL) even though the cockatiels have a much longer drug half-life and were given IM 5 mg/kg twice daily as opposed to once daily for the budgerigars [57,61]. This is likely attributable to the later first time point for data collection in the cockatiels (1.5 h) compared to the budgerigars (0.25 h). The first time point for data collection should be earlier so that a more accurate C_max_ value could be obtained. This would also allow for a more meaningful comparison between the C_max_ and T_max_ data with the other studies in which the first time points for data collection were either 0, 0.25, or 0.5 h.

The C_max_ values obtained after an IM 10 mg/kg single dose in the budgerigar [57], galah, scarlet macaw [62], golden eagle, great horned owl, red-tailed hawk [64], greater sandhill crane, and ring-necked pheasant [63] are very similar (approximately 35–37 μg/mL) even though they are very different species. However, the half-life of gentamicin for this dose is different and ranged from 0.53 to 2.74 h between the different species mentioned. Thus, C_max_ similarity is unlikely due to similarities in drug clearance. It is difficult to tell whether the C_max_ similarity is due to similar drug absorption rates because among the different species, T_max_ ranged from 0.25 to 1 h whereas the first time point for data collection ranged from 0 to 0.5 h.

### 7.2. Amikacin

The adverse effects and dosing recommendations of amikacin are presented in Table 8 and the PK data are presented in Table 9. Amikacin appears to be safe for exotic birds as no adverse effect was reported in any study.

Within the same species, amikacin appears to exhibit linear kinetics as the half-life is similar regardless of the dose or route of administration. The half-life of amikacin after being administered by the IM route is very similar among the different psittacine species (African grey parrot, blue-fronted amazon parrot; and cockatiel) and ranges from 0.97 h to 1.29 h regardless of weight. This may be due to inter-species similarities in drug clearance.

Most of the dosing regimens used did not result in a C_max_:MIC of at least 8–12 and hence may not be effective in eradicating an infection. Higher dose regimens should be tested in future studies.

## 8. Cephalosporins

Cephalosporins are bactericidal agents that are classified into different generations based on their antimicrobial activity. Third-generation cephalosporins are broad-spectrum antibiotics that are highly active against *Enterobacteriaceae* such as *Escherichia coli*, *Klebsiella* spp., and *Salmonella* spp. In humans, most cephalosporins are well absorbed after oral administration, primarily excreted by the kidneys, well-distributed throughout the body, and undergo minimal metabolism [32]. This review will focus on ceftiofur, a third-generation cephalosporin, as it is the most commonly used cephalosporin in birds [12,33]. Furthermore, all studies on cephalosporins in exotic avian species found in the literature are on ceftiofur except for one study [71]. Ceftiofur crystalline-free acid (CCFA) was used in all studies except for one in which ceftiofur sodium was used [72].

The adverse effects and dosing recommendations of ceftiofur are presented in Table 10 and the PK data are presented in Table 11. With the exception of mild tissue inflammation [73,74], no other adverse effect was observed [72,75,76,77,78].

The psittacine species (cockatiel and orange-winged amazon parrot (OWAP)) have shorter half-lives [72] than the other species tested. However, it cannot be deduced that ceftiofur has a shorter half-life in psittacine birds as current data are insufficient to accurately establish this relationship since only a few species have been studied. The ringneck dove is the only exotic bird species in which a high dose (50 mg/kg) was tested. However, no significant adverse effect was observed [74]. Thus, this high dose could potentially also be tested in other species.

All studies were single-dose studies involving long-acting ceftiofur. The T > MIC was at least 72 h in most species and a single dose may be adequate to eliminate an infection considering T > MIC only needs to be at least 40–50% of dosing intervals.

## 9. Tetracyclines

Tetracyclines are bacteriostatic agents which inhibit the growth of bacteria. They are mainly active against Gram-negative organisms. In humans, the absorption of most tetracyclines from the GI tract is incomplete and can be inhibited by the presence of divalent and trivalent cations. However, doxycycline has good oral absorption that is unaffected by the presence of food. Tetracyclines have a high volume of distribution and are predominantly eliminated by the kidneys, with the exception of doxycycline, which is excreted mostly unchanged in the bile and urine [32].

Avian chlamydiosis is a disease caused by the obligate intracellular Gram-negative bacterium *Chlamydia psittaci*. It usually affects psittacine birds such as budgerigars, parrots, and macaws. Doxycycline is the most commonly used and studied tetracycline in exotic birds [12,33]. It is also the drug of choice for the treatment of *Chlamydia psittaci* infections [3]. The adverse effects and dosing recommendations of doxycycline are presented in Table 12 and the PK data are presented in Table 13.

For IM administration, the commercial formulation of doxycycline (Vibravenos^®^) appears to be more effective (longer T > MIC) and safer [79,80] than the pharmacist-compounded formulations [80,81]. The 100 mg/mL pharmacist-compounded formulation did not result in plasma drug concentrations of at least 1 μg/mL in any of the three psittacine species that it was tested in. In contrast, the 75 mg/mL Vibravenos^®^ formulation sustained plasma concentrations above 1 μg/mL for at least 7 days in Goffin’s cockatoo and 18 h in OWAP. The difference in drug PK between these two formulations has been attributed to inconsistencies in the preparation and changes in rates of drug absorption that were dependent on the pH and concentration of the product. Because the pharmacist-compounded formulation is hard to compound and may lead to inconsistences between different batches, Vibravenos^®^ is preferred for IM injections.

In all the medicated food and water studies [81,82,83,84,85,86,87], plasma samples were collected very infrequently. For example, in one study, plasma samples were collected only four times in each bird throughout the 45-day treatment period [81]. This makes the determination of the T > MIC value difficult.

The common ostrich has a shorter half-life [88] than other bird species.

**Table 12 antibiotics-11-00741-t012:** Adverse effects and efficacy parameters of doxycycline in exotic avian species.

Dosage	Product	Species	MIC (μg/mL)	T > MIC	Adverse Effects	Ref.
**Intramuscular**						
15 mg/kg SD	Doxycycline hyclate powder	Common ostrich	1	≥1 h	Not reported.	[88]
100 mg/kg SD	Pharmacist compounded 75 mg/mL	Goffin’s cockatoo	1	≥7 days	Markedly higher AST, CK, LD. LD levels that were normal by 96 h.	[80]
		Orange-winged amazon parrot	1	~18 h	Markedly higher AST and CK.	
	Pharmacist compounded 100 mg/mL	Goffin’s cockatoo	1	0 h	Injection site abnormalities on day 7. Most birds had a firm 8–10 mm mass that resolved by 15–28 days after treatment.	
		Orange-winged amazon parrot	1	0 h	Injection site abnormalities on day 7. Most had a palpable 10–15 mm mass even at 38 days after injection.	
		Tinneh African grey parrot	1	0 h	Marked, moderate, and slight increase in CK, AST, and LD, respectively, on day 1. Values normal by day 7. All birds had palpable 10–12 mm injection site masses. By day 38, 50% of birds still had palpable 5–8 mm masses.	
	Vibravenos^®^ (20 mg/mL)	Orange-winged amazon parrot	1	5 days	None observed.	
100 mg/kg q10 days × 5 doses	Pharmacist compounded 100 mg/mL	Cockatiel	1	Not given; 5/35 samples were >1 μg/mL; samples collected at 0700 on days 7, 10, 20, 30, 40, 51	Mild-moderate swelling, bruising, and drug leakage from site of injection. Most local reactions were resolved within 10 days. However, muscle swelling, and a firm nodule that was present lasted several weeks.	[81]
100 mg/kg × 7 doses at intervals 7, 7, 7, 6, 6, 5 days	Vibravenos^®^ (20 mg/mL)	Houbara bustard	1	45 days	Reversible darkening of iris pigmentation. Moderate macroscopic changes at injection site, even after 7th injection. A small haemorrhage and drug leakage occasionally seen.	[79]
**Intravenous**						
15 mg/kg SD	Doxycycline hyclate powder	Common ostrich	1	≥12 h	Not reported.	[88]
**Subcutaneous**						
100 mg/kg SD	Pharmacist compounded 100 mg/mL	Tinneh African grey parrot	1	0	Injection site changes: yellow-stained skin, swelling, redness, 0.5 × 1 × 2 cm scab. Increased CK on day 1, normal by day 7. By day 38, scars began to form. Repeated injection may lead to the unacceptable sloughing of skin.	[80]
100 mg/kg × 7 doses at intervals 7, 7, 7, 6, 6, 5 days	Vibravenos^®^ (20 mg/mL)	Houbara bustard	1	45 days	Reversible darkening of iris pigmentation. Injection site showed slight irritation, sometimes in the form of a thickening of the skin or mild inflammation.	[79]
**Oral**						
300 mg/kg of pellets for 47 days	-	Cockatiel	1	Not given (all but 1 measured sample > 1 μg/mL) samples collected at 0830 on days 3, 7, 14, 21, 28, 35, 42	1 bird became markedly obese.	[82]
300 mg/kg of seeds for 42 days	-	Budgerigar	-	-	No notable adverse effects.	[84]
500 mg/kg of seed mixture for 45 days	-	Cockatiel	1	Not given (all samples > 1 μg/mL, except one sample of 0.82 μg/mL on day 35) samples collected at 0700 on days 3, 7, 15, 25, 35, 45	1 bird died on day 14.	[81]
1000 mg/kg of corn for 45 days	-	Blue-and-gold macaws, scarlet macaws	1	Not given (87% of samples were >1 μg/mL) samples collected on days 3, 15, 30, and 45)	None observed.	[86]
1000 mg/kg of mash for 45 days	-	Cockatiel	1	Not given (all samples > 1 μg/mL) samples collected at 0700 on days 3 and 7	Severe signs of toxicosis: reduced body weight, severe anorexia and lethargy, 1 bird died. Treatment terminated on day 3.	[81]
0, 50, 100, 200, 400 mg/L for 14 days	Medicated water	Budgerigar	-	Water containing ≤ 400 mg/L did not maintain plasma doxycycline concentrations of ≥1 μg/mL	No notable adverse effects.	[84]
280 mg/L for 45 days		Cockatiel	1	Not given (all but 1 sample on day 45 (0.92 μg/mL) >1 μg/mL) samples collected at 0700 on days 10, 20, 30, 45	Water consumption was significantly higher at the end of the trial. One bird consistently polyuric and polydipsic throughout trial, for reasons unknown.	[81]
400 mg/L for 30 days		Cockatiel	-	-	No clinically important adverse effects were associated with treatment.	[83]
400 mg/L for 7 days		Goffin’s cockatoo	1	Not given (all samples > 1 μg/mL) samples collected on days 3 and 7	Not reported.	[85]
		African grey parrot	1	0	Not reported.	
		Orange-winged amazon parrot	1	Not given (about 1 μg/mL in all samples) samples collected on days 3 and 7	Not reported.	
500 mg/L for 45 days		Beautiful, black-naped, Jambu and ring-necked fruit doves	1	Not given (64/96 (66%) of samples ≥1 μg/mL) samples collected at 1100 on days 3, 8, 14, 21, 35, 42	None observed.	[87]
800 mg/L for 7 days		African grey parrot	1	0	Not reported.	[85]
		Goffin’s cockatoo	1	Not given (all samples > 1 μg/mL) samples collected on days 3 and 7	Not reported.	
		Orange-winged amazon parrot	1	Not given (about 1 μg/mL in all samples) samples collected on days 3 and 7	Not reported.	
800 mg/L for 42 days		African grey parrot	1	Not given (>1 μg/mL in 73% of samples); samples collected at 0830 on days 4, 7, 14, 21, 28, 35, 42	None observed.	
		Goffin’s cockatoo	1	Not given (all samples > 1 μg/mL) samples collected at 0830 on days 4, 7, 14, 21, 28, 35, 42	AST and LD were elevated in 3 birds, 1 bird had high acid concentration, suggestive of mild hepatic damage. All parameters returned to normal within 7 days of treatment termination.	
830 mg/L for 45 days		Cockatiel	1	Not given (all samples > 1 μg/mL) samples collected at 0700 on days 10, 20, 30, 45.	None observed.	[81]
15 mg/kg SD	Stomach tube	Common ostrich	1	0	Not reported.	[88]
35 mg/kg q24 h for 21 days	Crop gavage	Cockatiel	1	Not given (samples taken on day 14 and 21 at 2–4 h post-injection were >1 μg/mL)	None observed.	[89]
35 mg/kg q24 h for 45 days	Crop gavage	Cockatiel	1	Not given (samples taken on day 14 and 21 at 2–4 h post-injection were >1 μg/mL)	None observed.	
50 mg/kg SD for 45 days	Oral gavage	Exotic Columbiformes	-	-	None observed.	[90]

**SD** = single dose; **MIC** = minimum inhibitory concentration (please refer to the original article for details on the specific pathogen being targeted); **T > MIC** = the duration plasma drug concentration exceeds MIC; **LD** = L-lactate dehydrogenase; **AST** = aspartate transaminase; **CK** = creatinine kinase.

**Table 13 antibiotics-11-00741-t013:** Pharmacokinetic values for doxycycline in exotic avian species.

Dosage	Product	Species	Weight (kg)	Half-Life (h)	C_max_ (μg/mL)	T_max_ (h)	Ref.
**Intramuscular**							
15 mg/kg SD	Doxycycline hyclate powder	Common ostrich	70–90	25.02 ± 3.98	1.35 ± 0.33	0.75 ± 0.18	[88]
100 mg/kg SD	Pharmacist compounded 75 mg/mL	Goffin’s cockatoo	0.281 ± 0.036	-	3.49 ± 0.18	-	[80]
		Orange-winged amazon parrot	0.416 ± 0.035	-	2.54 ± 0.38	-	
	Pharmacist compounded 100 mg/mL	Goffin’s cockatoo	0.281 ± 0.036	-	-	-	
		Orange-winged amazon parrot	0.416 ± 0.035	-	-	-	
		Tinneh African grey parrot	0.324 ± 0.023	-	-	-	
	Vibravenos^®^ (20 mg/mL)	Orange-winged amazon parrot	0.416 ± 0.035	74.2 ± 8.4	9.33 ± 0.82	3	
100 mg/kg q10 d × 5 doses	Pharmacist compounded 100 mg/mL	Cockatiel	0.082–0.126	-	-	-	[81]
100 mg/kg × 7 doses at intervals 7, 7, 7, 6, 6, 5 days	Vibravenos^®^ (2%)	Houbara bustard	0.985–1.765	IM1: 85.98 IM7: 77.12	IM1: 10.25 ^a^ IM7: 5.9 ^a^	IM1: 12 h IM7: 24 h	[79]
**Intravenous**							
15 mg/kg SD	Doxycycline hyclate powder	Common ostrich	70–90	-	-	-	[88]
**Subcutaneous**							
100 mg/kg SD	Pharmacist compounded 100 mg/mL	Tinneh African grey parrot	0.324 ± 0.023	-	-	-	[80]
100 mg/kg × 7 doses at intervals 7, 7, 7, 6, 6, 5 days	Vibravenos^®^ (2%)	Houbara bustard	0.985–1.765	SC1: 63.15 SC2: 58.93	SQ1: 6.75 ^a^ SQ7: 5.75 ^a^	SQ1: 12 SQ7: 24	[79]
**Oral**							
300 mg/kg of pellets for 47 days	-	Cockatiel	0.080–0.109	-	-	-	[82]
300 mg/kg of seeds for 42 days	-	Budgerigar	0.025–0.035 ^b^	-	-	-	[84]
500 mg/kg of seed mixture for 45 days	-	Cockatiel	0.080–0.109	-	-	-	[81]
1000 mg/kg of corn for 45 days	-	Blue-and-gold macaws, scarlet macaws	0.854–1.191	-	-	-	[86]
1000 mg/kg of mash for 45 days	-	Cockatiel	0.080–0.109	-	-	-	[81]
0, 50, 100, 200, 400 mg/L for 14 days	Medicated water	Budgerigar	0.025–0.035 ^b^	-	-	-	[84]
280 mg/L for 45 days		Cockatiel	0.082–0.126	-	-	-	[81]
400 mg/L for 7 days		African grey parrot	0.333 ± 0.020	-	-	-	[85]
		Goffin’s cockatoo	0.275 ± 0.031	-	-	-	
		Orange-winged amazon parrot	0.406 ± 0.033	-	-	-	
400 mg/L for 30 days		Cockatiel	0.08–0.125 ^b^	-	-	-	[83]
500 mg/L		Beautiful, black-naped, Jambu and ring-necked fruit doves	-	-	-	-	[87]
800 mg/L for 7 days		African grey parrot	0.333 ± 0.020	-	-	-	[85]
		Goffin’s cockatoo	0.275 ± 0.031	-	-	-	
		Orange-winged amazon parrot	0.406 ± 0.033	-	-	-	
800 mg/L for 42 days		African grey parrot	0.333 ± 0.020	-	-	-	
800 mg/L for 42 days		Goffin’s cockatoo	0.275 ± 0.031	-	-	-	
830 mg/L for 45 days		Cockatiel	0.082–0.126	-	-	-	[81]
PO 15 mg/kg SD	Stomach tube	Common ostrich	70–90	19.25 ± 2.53	0.30 ± 0.04	3.03 ± 0.48	[88]
PO 35 mg/kg q24 h for 21 days	Crop gavage	Cockatiel	0.08–0.125 ^b^	-	-	-	[89]
PO 35 mg/kg q24 h for 45 days	Crop gavage	Cockatiel	0.08–0.125 ^b^	-	-	-	
PO 50 mg/kg q24 h for 45 days	Oral gavage	Exotic Columbiformes	-	-	-	-	[90]

**SD** = single dose; **C_max_** = peak plasma drug concentration; **T_max_** = time to peak plasma drug concentration. ^**a**^: mg/L converted to μg/mL by dividing the original value by 1000 and then multiplying by 1000. ^**b**^: Mature budgerigars and cockatiels used in the study. Weight of bird obtained from https://lafeber.com/vet/content_types/information-sheet/?fwp_species=avian (accessed on 1 May 2022).

## 10. Recommendations for Practice

### 10.1. Doses Found for New Species

Our review found evidence from the literature for dosing of enrofloxacin and ceftiofur in some species that was not found in the Exotic Animal Formulary. For enrofloxacin, the recommended dosing regimen for “most species” in the Exotic Animal Formulary is PO or subcutaneous (SC) administration 15 mg/kg q12 h. Our critical review found dosing data for Caribbean flamingos, southern crested caracaras, and African penguins, species for which the formulary does not have specific dosing recommendations. The dosing regimen for “most species” in the formulary is higher than the PO or SC 15 mg/kg q24 h regimen recommended for Caribbean flamingos [48] and different from the PO pill/pill in fish 15 mg/kg q24 h regimen recommended for African penguins [42]. The various dosing regimens recommended for “raptors” in the formulary is different from the IV 7.5 mg/kg q24 h and IM 9.5 mg/kg q24 h regimens recommended for southern crested caracaras [45]. The common ostrich and greater rhea are also species for which the formulary does not have specific dosing recommendations. Our critical review found specific dosing regimens for the common ostrich, SC 15 mg/kg q12 h [35] and the greater rhea, IM 15 mg/kg [34]. These dosages are much higher than those recommended in the formulary for ratites: PO or SC 1.5–2.5 mg/kg q12 h and IM 5 mg/kg q12 h × 2 days [91].

For ceftiofur, our review found that IM 20 mg/kg is safe and effective for use in cattle egrets [75].

### 10.2. Doses That May Not Be Efficacious

#### 10.2.1. Enrofloxacin

The dosage regimens recommended in the Exotic Animal Formulary for ratites [91] may not be effective in common ostriches as a higher dose of enrofloxacin (15 mg/kg) than that recommended in the formulary was only sufficient to treat infections caused by highly susceptible organisms (MIC ≤ 0.136 μg/mL) [35]. This should certainly be ascertained through conducting a study on these birds using the dose recommended in the formulary.

#### 10.2.2. Doxycycline

The medicated mash (RBCO diet) regimen comprising 29% cooked white rice, 29% canned red kidney beans, 29% canned cooked corn, and 13% dry oatmeal, with 6.25 mg/kg brown sugar, resulted in severe toxicosis and death in the treated cockatiels as the low energy content of the mash led the cockatiels to consume a lot of the medicated mash to meet their daily energy requirement [81]. In the formulary, this exact regimen is recommended for large psittacine birds. In view of the serious adverse effects observed in the cockatiels, this regimen should be used with caution in other psittacine birds.

The dose range of the medicated water regimens recommended in the Exotic Animal Formulary is 130–800 mg/L [91]. Among the PK studies, the dose range which maintained plasma concentrations of at least 1 μg/mL is 280–830 mg/L [81,83,84,85,87]. The 130 mg/L medicated water regimen recommended for psittacine birds in the formulary may not be efficacious for budgerigars because it has been found that plasma concentrations of at least 1 μg/mL were not achieved in budgerigars given 0, 50, 100, 200, and 400 mg/L medicated water for 14 days [84]. The 830 mg/L medicated water for 45 days regimen for cockatiels was not mentioned in the formulary but could be recommended as it has been shown to be successful in maintaining plasma concentrations of at least 1 μg/mL for 45 days [81].

No specific dosing regimen of Vibravenos^®^ was available for OWAP in the formulary, even though one study found that a single IM dose of 100 mg/kg resulted in plasma concentrations above 1 μg/mL for at least 5 days in OWAP [80].

The PO 2–3.5 mg/kg q12 h regimen recommended for ratites in the formulary may not be successful if used in common ostriches because when a single dose of doxycycline PO 15 mg/kg was given to common ostriches, absolute bioavailability was only 5.03% and plasma concentrations did not reach at least 1 μg/mL at any of the time points measured [88].

### 10.3. Antibiotics That May Be Amendable for Dose Extrapolation

The antibiotics that are eliminated as primarily unchanged by the kidneys in humans (gentamicin and amikacin) seem to show less half-life variation across the different bird species than drugs that are primarily eliminated by other routes such as hepatic biotransformation (enrofloxacin, marbofloxacin, ceftiofur, and doxycycline). The half-life ranges of these antibiotics are presented in Table 14. This trend has been well reported in various review articles and has been hypothesized to be due to similarities in the glomerular filtration rate between the different bird species [13,92]. The implication of this trend is that drugs that are primarily eliminated by the kidneys may be more amenable to inter-bird-species dose extrapolation than other drugs. Although no adverse effects were reported in any of the amikacin studies included in this review, caution should be exercised when extrapolating amikacin dosage regimens to unstudied species as only five species of exotic birds have been studied to date. Furthermore, aminoglycosides are known to be nephrotoxic and have a narrow therapeutic index in humans [32]. Dose extrapolation should not be conducted using gentamicin at all as serious adverse effects have been reported in exotic birds.

## 11. Recommendations for Research

Firstly, the half-life of drugs seems to be shorter in the common ostrich than in the other exotic bird species. This trend was seen in enrofloxacin, marbofloxacin and doxycycline, the only antibiotics in this review that have been studied in ostriches. This trend is also seen in other drugs such as meloxicam and flunixin [94,95,96,97,98,99,100,101,102,103,104,105,106,107,108]. Other drugs have also been studied in the common ostrich [94,109,110,111], but no other studies have been conducted for other exotic avian species and therefore drug half-lives could not be compared. The reason for the shorter drug half-life observed in the common ostrich is unclear. This trend is counterintuitive to the observation that ostriches have abnormally low metabolic rates. Firstly, non-passerine birds have lower basal metabolic rates (BMRs) than passerine birds [112] but the BMR of the ostrich was found to be only 58% of the predicted non-passerine value [113]. Secondly, flightless birds have a lower BMR than flighted species [112], but when the BMRs of a group of 22 flightless bird species were plotted against their weights, the ostrich was an outlier as it had a lower BMR than expected for its weight [114]. Even though the active metabolic rate (AMR) of the common ostrich can be up to 28 times its BMR [113], it is unlikely that the common ostriches used in the study were very active. Furthermore, the AMR of birds when they are in flight can generally rise up to 23 times their BMR [115]. Future studies should be conducted to determine whether the trend of common ostriches having shorter drug half-life is also seen for drug classes other than antibiotics and non-steroidal anti-inflammatory agents. If this is the case, this trend could be used for dose extrapolation, especially since no adverse effects have been reported in common ostriches to date. Future studies could also investigate the anatomical and/or physiological difference between ostriches and other exotic birds such as its body fat and water composition that may impact the volume of distribution of the drug which could possibly explain this trend.

Secondly, studies need to be conducted on more exotic avian species for all the antibiotics listed in this review as only a few species have been studied for each antibiotic. Amoxicillin/clavulanate and trimethoprim/sulfamethoxazole are antibiotics which are commonly used in exotic avian veterinary practice because of their high palatability. In addition, amoxicillin/clavulanate is the drug of choice for the treatment of infections caused by *Yersinia tuberculosis* and *Pasteurella multocida* [33,116]. However, only two exotic avian species have been studied to date in total for amoxicillin/clavulanate [117,118].

Third, except for five studies [62,85,89,91,92], all included studies were conducted using birds which were not infected with bacteria. Future PK studies should use birds infected with bacteria as the state of health can affect the disposition of drugs.

Lastly, most current literature has only reported total concentrations instead of plasma concentrations. Future studies should include the measurement of plasma protein binding which may show significant interspecific variabilities as only the free antibiotic concentrations are clinically relevant.

## 12. Conclusions

Enrofloxacin, marbofloxacin, amikacin, and ceftiofur are generally safe for use in exotic birds as either no or only mild reversible adverse effects were reported. In contrast, gentamicin and doxycycline are less safe for use as serious adverse effects have been reported in some studies. In avian veterinary practice, antibiotic regimens used are typically obtained from the Exotic Animal Formulary. Our review found specific dosage recommendations for some species for which no specific dosage recommendations are provided in the formulary. Specific dosing regimens of enrofloxacin for Caribbean flamingos, southern crested caracaras, African penguins, greater rheas, and common ostriches were found. Specific dosing regimens of ceftiofur in cattle egrets and doxycycline in cockatiels were also found. These regimens could be used in practice. Amikacin appears to be the only antibiotic in this review that could be amenable to dosage extrapolation as it is generally safe and showed little PK variation across species. However, caution should be exercised as only a few exotic bird species were studied. Common ostriches seem to be unique in having shorter drug half-lives than other exotic birds.

## Figures and Tables

**Table 1 antibiotics-11-00741-t001:** Common poultry species included in this critical review.

English Name of Species	Scientific Name of Species
Red junglefowl	*Gallus gallus*
Mallard duck	*Anas platyrhynchos*
Muscovy duck	*Cairina moschata*
Wild turkey	*Meleagris gallopavo*
Greylag goose	*Anser anser*
Swan goose	*Anser cygnoides*

**Table 8 antibiotics-11-00741-t008:** Adverse effects, efficacy parameters and dosage recommendations of amikacin in exotic avian species.

Dosage	Species	MIC (μg/mL)	C_max_:MIC	Adverse Effects	Dosage Recommended by Authors	Ref.
**Intramuscular**						
5 mg/kg SD	African grey parrot	≤8	≥1.4	Not reported	IV or IM 10–20 mg/kg q8–12 h, depending on MIC	[67]
10 mg/kg SD	African grey parrot	≤8	≥2.64	Not reported	10–20 mg/kg IV or IM q8–12 h, depending on MIC	
15 mg/kg SD	Blue-fronted amazon parrot	16	2.38	Not reported	IV 15 mg/kg q8 h or IM 15 mg/kg q12 h	[68]
15 mg/kg q12 h × 3 d	Cockatiel	-	-	Weight-loss of 8.9% due to handling stress	IM 15–20 mg/kg either bd or tds for infections caused by susceptible bacteria	[61]
20 mg/kg SD	African grey parrot	≤8	≥4.09	Not reported	10–20 mg/kg IV or IM q8–12 h, depending on MIC	[67]
	Red-tailed hawks	0.5–8	~20 for most pathogens tested	None observed	IM 15–20 mg/kg/day either as a single dose or divided into 2 or 3 doses	[69]
**Intravenous**						
5 mg/kg SD	African grey parrot	≤8	≥3.8	Not reported	10–20 mg/kg IV or IM q8–12 h, depending on MIC	[67]
7.2 ± 0.12 mg/kg SD	Emu	8	≥4	Not reported	IV 7.2 ± 0.12 mg/kg q24 h	[70]
10 mg/kg SD	African grey parrot	≤8	≥11.1	Not reported	10–20 mg/kg IV or IM q8–12 h, depending on MIC	[67]
15 mg/kg SD	Blue-fronted amazon parrot	16	6.25	Not reported	15 mg/kg q8 h IV or 15 mg/kg q12 h IM	[68]
20 mg/kg SD	African grey parrot	≤8	≥12.5	Not reported	10–20 mg/kg IV or IM q8–12 h, depending on MIC	[67]

**SD** = single dose; **MIC** = minimum inhibitory concentration (please refer to the original article for details on the specific pathogen being targeted); **C_max_** = peak plasma drug concentration; **T_max_** = time to peak plasma drug concentration; **C_max_:MIC** = ratio of peak plasma drug concentration to minimum inhibitory concentration.

**Table 9 antibiotics-11-00741-t009:** Pharmacokinetic values for amikacin in exotic avian species.

Dosage	Species	Weight (kg)	Half-Life (h)	C_max_ (μg/mL)	T_max_ (h)	V (L/kg)	Clearance (mL/kg/h)	AUC (μg·h/mL)	Ref.
**Intramuscular**									
5 mg/kg SD	African grey parrot	0.418–0.559	1.08	11.2 ^c^	0.25	0.34	191	-	[67]
10 mg/kg SD	African grey parrot	0.418–0.559	1.04	21.1 ^c^	0.75	0.39	232	-	
15 mg/kg	Blue-fronted amazon parrot	0.270–0.410	1.08	38	0.283	-	-	87.5 ^b^	[68]
15 mg/kg q12 h × 3 days	Cockatiel	0.104	1.29	27.3 ± 6.9	1	-	-	-	[61]
20 mg/kg SD	African grey parrot	0.418–0.559	0.97	32.7 ^c^	0.75	0.47	217	-	[67]
	Red-tailed hawks	0.900–1.825	2.02 ± 0.63	65 ± 12	0.5–0.75	0.28 ± 0.03	-	207 ± 46	[69]
**Intravenous**									
5 mg/kg SD	African grey parrot	0.418–0.559	1.06	30.4 ^c^	0.0833	0.23	188	-	[67]
7.2 ± 0.12 mg/kg SD	Emu	25–26	0.87	≥32.0	-	0.18	30 ^a^	269.66	[70]
10 mg/kg SD	African grey parrot	0.418–0.559	0.9	88.8 ^c^	0.0833	0.12	142	-	[67]
15 mg/kg SD	Blue-fronted amazon parrot	0.270–0.410	0.483	100	-	-	-	69.7 ^b^	[68]
20 mg/kg SD	African grey parrot	0.418–0.559	1.34	99.8 ^c^	0.0833	0.31	229	-	[67]

**SD** = single dose; **V** = apparent volume of distribution; **AUC** = area under the plasma concentration versus time curve. ^**a**^: Units converted from L/h/kg to mL/kg/h by multiplying the original value reported by 1000. ^**b**^: Units converted from μg/mL/min to μg·h/mL by dividing the original value by 60. ^**c**^: C_max_ and T_max_ not stated by the authors of the study. C_max_ assumed to be the highest plasma drug concentration measured and T_max_ assumed to be the time point for data collection at which the highest plasma drug concentration was measured.

**Table 10 antibiotics-11-00741-t010:** Adverse effects, efficacy parameters, and dosage recommendations of ceftiofur in exotic avian species.

Dosage	Species	MIC (μg/mL)	T > MIC (h)	Adverse Effects	Dose Recommended by Study	Ref.
**Intramuscular**						
10 mg/kg SD	American black duck	1 and 4	123 and 73.3	- ^b^	IM 10 mg/kg q3 d for future studies	[78]
	American flamingo	1	72 h in 100% of birds, 96 h in 82% of birds, 144 h in 18% of birds	Slight weight loss due to manual restraint	-	[76]
	Cockatiel	1	At least 4 h	None observed	IM 10 mg/kg q4 h	[72]
	Helmeted guineafowl	1	At least 56 h in all birds and for 72 h in 2 birds	None observed		[77]
	Orange-winged amazon parrot	1	At least 8 h	None observed	IM 10 mg/kg q8–12 h	[72]
	Red-tailed hawk	4 ^a^	36 h	Little to no muscle inflammation	-	[73]
20 mg/kg SD	Cattle egret	1	72 h in all birds, 96 h in 50% of birds	Not reported	-	[75]
	Red-tailed hawk	4 ^a^	96 h	Little to no muscle inflammation	-	[73]
50 mg/kg SD	Ringneck dove	1	108 h	Very mild tissue inflammation at injection site, appears to be safe	-	[74]
**Subcutaneous**						
10 mg/kg SD	American flamingo	1	72 h and 96 h, respectively, for the 2 birds	Slight weight loss due to manual restraint	-	[76]

**SD** = single dose; **MIC** = minimum inhibitory concentration (please refer to the original article for details on the specific pathogen being targeted); **T > MIC** = the duration plasma drug concentration exceeds MIC. ^**a**^: MIC used was 1 μg/mL, but target plasma concentration used to evaluate T > MIC is 4 μg/mL. ^**b**^: no information as full text was not available.

**Table 11 antibiotics-11-00741-t011:** Pharmacokinetic values for ceftiofur in exotic avian species.

Dosage	Species	Weight (kg)	C_max_ (μg/mL)	T_max_ (h)	Half-Life (h)	AUC (μg·h/mL)	Clearance (mL/kg/min)	Ref.
**Intramuscular**								
10 mg/kg SD	American black duck	0.720–1.640 ^a^	13.1	24	32	783	-	[78]
	American flamingo	≥2.4	7.49 ± 1.9	27 ± 13	39.9 ± 9.7	525 ± 123	-	[76]
	Cockatiel	0.091 ± 0.008	5.25	-	2.5	14.7	11.3	[72]
	Helmeted guineafowl	1–1.6	5.26	19.3	29.0 ± 4.93	306 ± 69.3	-	[77]
	Orange-winged amazon parrot	0.393 ± 0.032	10.99	-	7.9	43.8	3.8	[72]
	Red-tailed hawk	0.690–1.46 ^a^	6.8	6.4	29	-	-	[73]
20 mg/kg SD	Cattle egret	0.34	16.22 ± 5.11	3.20 ± 2.6	37.92 ± 7.49	451.30 ± 141.0	-	[75]
	Red-tailed hawk	0.690–1.46 ^a^	15.1	6.7	50	-	-	[73]
50 mg/kg SD	Ringneck dove	0.156 ^a^	-	-	-	-	-	[74]
**Subcutaneous**								
10 mg/kg SD	American flamingo	≥2.4	6 and 4.5 in 2 different birds	48 and 12 in 2 different birds	-	-	-	[76]

**SD** = single dose; **C_max_** = peak plasma drug concentration; **T_max_** = time to peak plasma drug concentration; **AUC** = area under the plasma concentration versus time curve. ^**a**^: weight of American black duck, helmeted guineafowl, red-tailed hawk, and ringneck dove were obtained from https://www.ducks.org/hunting/waterfowl-id/american-black-duck (accessed on 1 May 2022), https://www.marylandzoo.org/animal/helmeted-guinea-fowl/ (accessed on 1 May 2022), https://www.allaboutbirds.org/guide/Red-tailed_Hawk/overview (accessed on 1 May 2022), and https://sharon.audubon.org/our-resident-ring-necked-dove (accessed on 1 May 2022), respectively, as full text of article was not available.

**Table 14 antibiotics-11-00741-t014:** Half-life ranges of enrofloxacin, marbofloxacin, gentamicin, amikacin, ceftiofur, and doxycycline in exotic avian species.

Antibiotic	Half-Life Range (h)	No. of Species	Elimination of Antibiotic	Ref.
Enrofloxacin	0.723–82.5	11	Metabolized to ciprofloxacin (in birds)	[36]
Marbofloxacin	1.61–15.03	4	Metabolized by the liver (in birds)	[93]
Gentamicin	0.53–3.36	10	Excreted primarily unchanged by the kidneys (in humans)	[32]
Amikacin	0.483–2.63	3	Excreted primarily unchanged by the kidneys (in humans)	[32]
Ceftiofur	2.5–50	8	Rapidly metabolized to desfuroylceftiofur metabolites (in birds)	[77]
Doxycycline	21.04–85.98	3	Excreted mostly unchanged in the bile and urine (in humans)	[32]

## Data Availability

Not applicable.

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
