# Peer review of "A Critical Review of the Pharmacokinetics, Pharmacodynamics, and Safety Data of Antibiotics in Avian Species"

_antibiotics, 2022, doi:10.3390/antibiotics11060741_

Round 1

Reviewer 1 Report

In this review, the authors aim to provide literature information on the pharmacokinetics, pharmacodynamics and safety data of antibiotics in different exotic avian species.

The subject of the review is original and falls within the scope of the journal. Moreover, I think this review provides valuable data for scientific literature and the readers of the journal. 

  1. Title: “A critical review on the pharmacokinetics, pharmacodynamics and safety of antibacterial drugs used in different exotic avian species”
  2. Considering the data in the table related to PK values and the species studied, the discussion parts for each drug (Marbofloxacin, Amikacin, Ceftiofur, Doxycycline) are extremely limited. In addition, pharmacodynamic data has not been adequately discussed in the manuscript. The pharmacodynamics and safety of antibiotics used in exotic avian species should be discussed separately.
  3. Line 71: “Antibiotics can be classified into two groups based on their PD properties”. But lines 81-82: “The third group consists …”?. Antibiotics can be classified according to their chemical structure, mechanism of action or spectrum of activity. The classification of antibiotics should be revised according. Moreover, general information on antibiotics such as mode of action, the spectrum of activity (important bacterial infections in exotic avian species), administration routes, and pharmaceutic forms should be given in the introduction of the manuscripts.
  4. Table 1: This table was not cited in the text and should be removed since it does not make any sense!
  5. Line 124-125: “The adverse effects and dosing recommendations of enrofloxacin in various avian species are presented in Table 2 and the PK data are presented in Table 3”.
  6. Lines 175-185: The AUC and Tmax values of both species should be given here.
  7. Table 2 and the other similar tables: The MIC values should be described. MIC values are calculated against specific pathogens and these pathogens should be given in tables.
  8. Line 251: Should be “… belong to Accipitriformes species”.
  9. Line 255: Should be “… belongs to Psittaciformes species”.
  10. Line 255: “Thus, caution has to be taken …”.
  11. Line 264: “Table 5. Pharmacokinetic … “.
  12. P68, lines 17-22: “Our critical review found dosing data…” and Line 32: “Our review found that IM 20mg/kg is safe and effective for use in cattle egrets”. How were these assessments made?
  13. Reference list: Please check the consistency of the reference list for the guidelines of the journal
  14. Supplementary Materials: The link given here does not work!
  15. The other antibiotics studied in exotic avian species (some are given below) should be provided in a table and discussed in the manuscript.

Houben, R., Antonissen, G., Croubels, S., De Backer, P., & Devreese, M. (2016). Pharmacokinetics of drugs in avian species and the applications and limitations of dose extrapolation. Vlaams Diergeneeskundig Tijdschrift, 85(3).

Osofsky, A., Tell, L. A., Kass, P. H., Wetzlich, S. E., Nugent‐Deal, J., & Craigmill, A. L. (2005). Investigation of Japanese quail (Coturnix japonica) as a pharmacokinetic model for cockatiels (Nymphicus hollandicus) and Poicephalus parrots via comparison of the pharmacokinetics of a single intravenous injection of oxytetracycline hydrochloride. Journal of veterinary pharmacology and therapeutics, 28(6), 505-513.

Teare, J. A., Schwark, W. S., Shin, S. J., & Graham, D. L. (1985). Pharmacokinetics of a long-acting oxytetracycline preparation in ring-necked pheasants, great horned owls, and Amazon parrots. American Journal of Veterinary Research, 46(12), 2639-2643.

Koc, F., Uney, K., Ozturk, M., Kadioglu, Y., & Atila, A. (2009). Pharmacokinetics of florfenicol in the plasma of Japanese quail. New Zealand Veterinary Journal, 57(6), 388-391.

Ismail, M., & El-Kattan, Y. A. (2009). Comparative pharmacokinetics of florfenicol in the chicken, pigeon and quail. British poultry science, 50(1), 144-149.

Lenarduzzi, T., Langston, C., & Ross, M. K. (2011). Pharmacokinetics of clindamycin administered orally to pigeons. Journal of Avian Medicine and Surgery, 25(4), 259-265.

Carpenter, J. W., Olsen, J. H., Randle-Port, M., Koch, D. E., Isaza, R., & Hunter, R. P. (2005). Pharmacokinetics of azithromycin in the blue and gold macaw (Ara ararauna) after intravenous and oral administration. Journal of Zoo and Wildlife Medicine, 36(4), 606-609.

Ensley, P. K., & Janssen, D. L. (1981). A Preliminary Study Comparing the Pharmacokinetics of Ampicillin Given Orally and Intramuscularly to Psittacines: Amazon Parrots (Amazond spp.) and Blue-Naped Parrots (Tanygnathus lucionensis). The Journal of Zoo Animal Medicine, 12(2), 42-47.

Kilburn, J. J., Cox, S. K., & Backues, K. A. (2016). Pharmacokinetics of ceftiofur crystalline free acid, a long-acting cephalosporin, in American flamingos (Phoenicopterus ruber). Journal of Zoo and Wildlife Medicine, 47(2), 457-462.

Schmitt, T. L., Nollens, H. H., Simeone, C. A., & Papich, M. G. (2019). Population Pharmacokinetics of Danofloxacin After Single Intramuscular Dose Administration in California Brown Pelicans (Pelecanus occidentalis californicus). Journal of Avian Medicine and Surgery, 33(4), 361-368.

Tekeli, I. O., Turk, E., Durna Corum, D., Corum, O., Kirgiz, F. C., & Uney, K. (2020). Pharmacokinetics, bioavailability and tissue residues of doxycycline in Japanese quails (Coturnix coturnix japonica) after oral administration. Food Additives & Contaminants: Part A, 37(12), 2082-2092.

Yang, F., Wang, H., Song, Z. W., Yu, M. L., Zhang, M., Wang, X. D., & Kang, T. J. (2021). Pharmacokinetics of Ceftiofur Sodium in Black-bone Silky Fowl after One Single Intravenous and Intramuscular Injection. Indian Journal of Animal Research, 1, 5.

Dimitrova, D. J., Haritova, A. M., Dinev, T. D., Moutafchieva, R. G., & Lashev, L. D. (2014). Comparative pharmacokinetics of danofloxacin in common pheasants, guinea fowls and Japanese quails after intravenous and oral administration. British poultry science, 55(1), 120-125.

Aboubakr, M. (2012). Pharmacokinetics of levofloxacin in Japanese quails (Coturnix japonica) following intravenous and oral administration. British poultry science, 53(6), 784-789.

Author Response

Reviewer 1

  1. Title: “A critical review on the pharmacokinetics, pharmacodynamics and safety of antibacterial drugs used in different exotic avian species”

Original title: A critical review of the pharmacokinetic, pharmacodynamic, and safety data of antibiotic dosing regimens used in different exotic avian species”.

According to the Alliance for Prudent Use of Antibiotics (APUA) website: “Antibacterials are now most commonly described as agents used to disinfect surfaces and eliminate potentially harmful bacteria. Unlike antibiotics, they are not used as medicines for humans or animals, but are found in products such as soaps, detergents, health and skincare products and household cleaners.”

As such, we have amended the title to “A critical review on the pharmacokinetics, pharmacodynamics and safety of antibiotics used in avian species”.

Reference: https://apua.org/antibacterial-agents

  1. Considering the data in the table related to PK values and the species studied, the discussion parts for each drug (Marbofloxacin, Amikacin, Ceftiofur, Doxycycline) are extremely limited. In addition, pharmacodynamic data has not been adequately discussed in the manuscript. The pharmacodynamics and safety of antibiotics used in exotic avian species should be discussed separately.

More than 90% of the included articles were pharmacokinetic studies conducted in healthy birds. The measure of efficacy was a prediction based on concentrations achieved. Therefore, it is challenging to expand the discussion on the pharmacodynamics and safety of the antibiotics based on the study design of the original articles.

  1. Line 71: “Antibiotics can be classified into two groups based on their PD properties”. But lines 81-82: “The third group consists...”?. Antibiotics can be classified according to their chemical structure, mechanism of action or spectrum of activity. The classification of antibiotics should be revised according. Moreover, general information on antibiotics such as mode of action, the spectrum of activity (important bacterial infections in  exotic avian species), administration routes, and pharmaceutic forms should be given in the introduction of the manuscripts.

Section 2 (which has now been renumbered as Section 4) introduces the broad pharmacodynamic classification of antibiotics. We apologize for the confusing use of “two groups” which has now been removed. The specific mechanism of action (bactericidal or bacteriostatic) and spectrum of activity of the various classes of antibiotics (fluoroquinolones, aminoglycosides, cephalosporins, and tetracyclines) discussed in the review are stated at the beginning of each respective sections. The avian bacterial diseases these antibiotics are usually used to treat are also stated in the same sections.

  1. Table 1: This table was not cited in the text and should be removed since it does not make any sense!

The table was incorrectly cited as Table 2 in-text. This has now been corrected and the Table title has been revised to provide more clarity. This table is important in defining the scope of avian species included in this review.

  1. Line 124-125: “The adverse effects and dosing recommendations of enrofloxacin in various avian species are presented in Table 2 and the PK data are presented in Table 3”.

We are unclear what this feedback entails.  

  1. Lines 175-185: The AUC and Tmax values of both species should be given here.

Amended.

  1. Table 2 and the other similar tables: The MIC values should be described. MIC values are calculated against specific pathogens and these pathogens should be given in tables.

We agree that inclusion of pathogens in the studies would make interpretation of MIC values more meaningful. However, as mentioned above, more than 90% of the studies were pharmacokinetic studies using healthy birds. The concentration achieved was used to predict efficacy based on whether target MIC would have been achieved or not. The MIC values were typically obtained from other studies by the authors and if the antibiotic targets different microorganism, the highest MIC would be chosen as a surrogate on whether efficacy would be achieved with the concentrations observed. We have added Section 5 which provides this background now. In addition, we have included in the footnote of relevant tables with MIC values to advise readers to refer to the original articles for more details.    

  1. Line 251: Should be “... belong to Accipitriformes species”.

Accipitriformes is an order of birds.

  1. Line 255: Should be “... belongs to Psittaciformes species”.

Psittaciformes is an order of birds.

  1. Line 255: “Thus, caution has to be taken ...”.

Amended.

  1. Line 264: “Table 5. Pharmacokinetic ... “.

Capitalized “pharmacokinetic”.

  1. P68, lines 17-22: “Our critical review found dosing data...” and Line 32: “Our review found that IM 20mg/kg is safe and effective for use in cattle egrets”. How were these assessments made?

 “A dose of 20 mg/kg was administered intramuscularly to 18 birds and blood samples were collected via jugular venipuncture at 1, 2, 4, 8, 12, 24, 48, 72, 96, 120, 144, 168, 192, 216, and 240 hours after CCFA administration. Plasma concentrations of ceftiofur free acid equivalents (CFAEs) were measured via high-performance liquid chromatography. The minimum inhibitory concentration (MIC) of 1 lg/mL was reached by 1 hour after administration and remained higher than the MIC for at least 72 hours in all birds.”

“The reported MIC90 of ceftiofur for numerous avian bacterial pathogens is 1 lg/mL. This includes Enterobacter species, Pasteurella spp, Staphylococcus aureus, Proteus species, Klebsiella species, Salmonella species, and Escherichia coli.”

For time-dependent antibiotics such as ceftiofur, the amount of time that serum concentrations remain above the MIC is the most significant parameter for determining efficacy of the antibiotic. No adverse effects were observed in any bird. Thus, the IM 20mg/kg ceftiofur dosing regimen is likely to be safe and effective for cattle egrets.

Reference: Waldoch JA, Cox SK, Armstrong DL. Pharmacokinetics of a Single Intramuscular Injection of Long-Acting Ceftiofur Crystalline-Free Acid in Cattle Egrets (Bubulcus ibis). J Avian Med Surg. 2017 Dec;31(4):314-318. doi: 10.1647/2016-222. PMID: 29327960

  1. Reference list: Please check the consistency of the reference list for the guidelines of the journal

Updated.

  1. Supplementary Materials: The link given here does not work!

Table S1 is appended at the end of the same file (not as a separate file).

  1. The other antibiotics studied in exotic avian species (some are given below) should be provided in a table and discussed in the manuscript.

We thank the reviewer for this feedback and suggestion. At present, the scope of the review is already voluminous. We feel that adding more antibiotics with limited data available may not be very meaningful as these studies are likely cited in the Exotic Animal Formulary already.  

Reviewer 2 Report

The current manuscript deal with pharmacokinetic, pharmacodynamic, and safety data of antibiotic dosing regimens in different exotic avian species.

The paper presents interesting informations but the way in which they are presented is quite dissapointing, the authors should be paying more attention to this aspect also. For example, the page no of the manuscript is missing, so is quite hard to indicate the exact page where corrections are nedeed, and also at some point the right page numbering is starting from 1 again. 

introduction part row 52 to 55. I don t see the point of this sentence. It is obvious that the antibiotics have different effect among different avian species. My recomendation is to remove this sentence or to rephrase it. 

3 rd point from the aim of this study, is not very clear for me.

In Tables, the authors present the use of antibiotics on quails. I have to remind them that Quails are also consumed by humans (egg and meat industries). Since 2006, in the European Union the use of antibiotics was banned in animals designated for human consumption, due to increased drug resistance against certain bacteria. For this reason, I think that all the informations regarding the use of antibiotics on quails should be removed, or, please mention somwhere if the use of antibiotics in this particular birds is practiced only in certain countries. Also, there are no conclusions or recommendations about this birds. Please explain this important aspect.

It is not very clear for me where this antibiotics are used! In what countries? It is and worldwide study, or is based on studies from some particular country? I this it is worth mentioning this aspect also. 

The supplementary materials are unavailable according to the link provided in the manuscript. 

The taxonomic names of bacteria species should be italicised. 

Table 11. Remove respectively from the subcutaneos section of american flamingo .

Section 7.1.2. is about what? This information is missing. 

Sections 7.1.1. and 7.2.1. are confusing. I suggest to merge them in one section. 

Paragraph before the conclusion. I dont see the point, Please rephrase it, it is not very clear. 

Remove the Acknowledgments section if it is not applicable. 

Why the healthy animal have been treated with antibiotics? 

Please explain why this animals, used as pets needs to be given antibiotics? It is for prevention? Or why? 

The reference part it is not as per journal indications formated. 

Author Response

Reviewer 2

The current manuscript deal with pharmacokinetic, pharmacodynamic, and safety data of antibiotic dosing regimens in different exotic avian species. The paper presents interesting information but the way in which they are presented is quite disappointing, the authors should be paying more attention to this aspect also. For example, the page no of the manuscript is missing, so is quite hard to indicate the exact page where corrections are needed, and also at some point the right page numbering is starting from 1 again.

Error with page numbering have been rectified.

Introduction part row 52 to 55. I don t see the point of this sentence. It is obvious that the antibiotics have different effect among different avian species. My recommendation is to remove this sentence or to rephrase it.

This part has been removed.

3rd point from the aim of this study, is not very clear for me.

Aim has been elaborated.

In Tables, the authors present the use of antibiotics on quails. I have to remind them that Quails are also consumed by humans (egg and meat industries). Since 2006, in the European Union the use of antibiotics was banned in animals designated for human consumption, due to increased drug resistance against certain bacteria. For this reason, I think that all the information regarding the use of antibiotics on quails should be removed, or, please  mention somewhere if the use of antibiotics in this particular bird is practiced only in certain countries. Also, there are no conclusions or recommendations about this bird. Please explain this important aspect. It is not very clear for me where this antibiotics are used! In what countries? It is and worldwide study, or is based on studies from some particular country? I this it is worth mentioning this aspect also.

A systematic review-like approach was adopted in searching for the relevant articles based on the search concept of the specific antibiotic name and avian species in general (excluding the poultry species listed in Table 1). Therefore, the articles that were retrieved is from studies conducted worldwide. We acknowledge that Japanese quail is a common poultry species and have removed references relating to this species throughout the manuscript.

The supplementary materials are unavailable according to the link provided in the manuscript.

Table S1 is appended at the end of the same file (not as a separate file).

The taxonomic names of bacteria species should be italicised.

Amended.

Table 11. Remove respectively from the subcutaneous section of American flamingo .

Amended.

Section 7.1.2. is about what? This information is missing.

Amended.

Sections 7.1.1. and 7.2.1. are confusing. I suggest to merge them in one section.

Amended.

Paragraph before the conclusion. I don’t see the point, Please rephrase it, it is not very clear.

Amended.

Remove the Acknowledgments section if it is not applicable.

Removed.

Why the healthy animal have been treated with antibiotics?

Have clarified that these are PK studies i.e. the disposition of the drugs were investigated using healthy animals.

Please explain why this animals, used as pets needs to be given antibiotics? It is for prevention? Or why?

The antibiotics could be used for treatment of infection or prophylaxis post-surgery. If this comment is in relation to the fact that the animals are healthy, it is because they were pharmacokinetic studies.

The reference part it is not as per journal indications formatted.

Updated.

Reviewer 3 Report

The title of this review is attractive but its content is rather disappointing. It is rather a series of  tables compiling figures  reported in the literature without any real critical analysis of what has been  published. A critical review in this field requires a high level of expertise in pharmacokinetics (PK)  and also in pharmacodynamics. This is all the more important  since the primary literature on the PK  of exotic birds is often generated by non-specialists and the risk is to peddle in this review results  that are  more or less flawed or poorly interpreted (see my specific comments).

Moreover, I believe  that as a preliminary, the authors should recall some basic notions on the physiological, metabolic and behavioral specificities of exotic birds having possible consequences on drugs’ PK. They should explain whether the particularities reported in poultry, in particular on the absorption and elimination of drugs (1), (2) are also found or not in exotic birds. For example, does the existence  of a reptlian-type kidney with the possibility or not associated to  a first-pass effect for drugs administered intramuscularly in the thigh? if yes, this  require to qualify in this  review the site of IM administration because bioavailability could be very different depending of the IM site of administration. Similarly, behavioral particularities such as the fact that birds (at least poultry) only drink during the lighting period and that water consumption is dependent of the ambient temperature should be reported and discussed.  This is  because it is a problem for the oral route of administration when one wants to treat an infection   with a time-dependent antibiotic. They could learn from Vermelen 's excellent article (1) . It is also from the experience of this referee that certain PK concepts are not well understood by non-specialist  and the Authors must explain them.

Specific comments

Line 41-47: the Authors recommend making Antimicrobial Susceptibility Testing (AST) which here is not welcome. Indeed the Clinical Breakpoints (CBP) which are defined by major international organizations such as EUCAST, CLSI, USCAST, etc. are defined for each species and for well-defined dosages. However, these dosages are unknown for exotic birds! As such, the interpretation of an AST by taking CBP from another species can be very misleading. Rather, the authors could give for the main pathogens of interest in exotic birds the corresponding ECOFFs ( i.e. the epidemiological cut-off) which are specific to a given bacterial species and which have the same value in all animal species (see these ECOFFs on the EUCAST site: https://www.eucast.org/mic_distributions_and_ecoffs/). And it is the ECOFFs that can be used to interpret an AST in exotic birds in the absence of CBP. This is what is recommended by VetCAST, the veterinary subgroup of EUCAST. By following this approach, the authors could compare the plasma concentrations obtained with these ECOFFs (MIC of wild population) and thus make a first judgment on the validity of the dosages currently recommended for exotic birds (see the EUCAST/VetCAST website and the associated publications https://www.eucast.org/ast_of_veterinary_pathogens/ ).

Another important point that is not addressed in this paper is that all PK/PD indices must be calculated in terms of free, not total plasma concentrations (3) . I am very concerned that this information on plasma protein binding of antimicrobials  is not available and the Authors must discuss it, especially in their last part entitled “ recommendation for research ”.

Line 45, 74 …: PK/PD index are not to compute efficacy but only to predict efficacy. Furthermore, the Cmax/MIC index is no longer used either by EUCAST or by USCAST. According to USCAST, “Given the transient nature of Cmax and the ability to estimate AUC with greater precision than Cmax, AUC:MIC rather than Cmax:MIC ratio targets for efficacy were characterized and considered for PK-PD target attainment analyses, see https:/ /app.box.com/s/1hxc8inf8u3rranwmk3efx48upvwt0ww )

Line 85: it is true that the optimal target is unknown but a target of 13 h is very unlikely. Indeed , it is poorly understood that a target of 24 h just means that the average free plasma concentration over 24h and in steady-state conditions, is equal to the MIC and a target of 13 h mean that efficacy would be achieved with an average plasma concentration equal to only half the MIC!

Line 104-108: The Authors reported different PK  parameters and variables without ranking them. Actually, the most important of the PK parameters is the plasmatic clearance whereas the AUC is only a hybrid variable depending on  dose, clearance and bioavailability for the extravascular routes of administration. Authors needs to precisely qualify reported  PK parameters and their interpretation. Just taking the volume of distribution (Vd)  as an example, the Authors claim to report “the” volume of distribution when there are actually three volumes of distribution that are routinely reported in the literature (Vc, Vss and Varea) with definitions and interpretations that are different (4) ; moreover, for the extravascular routes, what is calculated is a V/F, i.e. an apparent volume of distribution which depends on the bioavailability (symbol F, not BA). This V/F is therefore not a parameter but a contextual variable with no interest in being reported in this review. Let's just take as an example the volume of distribution of enrofloxacin in Houbara bustard. Reported values for the IV, IM and oral routes  are 2.98, 3.18 and 5.12 L/Kg. In fact, the only relevant value (because it corresponds to a primary parameter) is that of the IV route, the other values being variables, not parameters  which depend on F. this explains the greater value of Vd for the oral route because its bioavailability is the lowest.

Line 132: I see no evidence of difference  in the half-life times of enrofloxacin in common ostrich in table 3. More importantly, the whole discussion on the interpretation of differences in half-life needs to be reconsidered. Authors should realize that the half-life is a hybrid parameter that depends on clearance and volume of distribution (here varea ) and that the first level of interpretation to explain observed differences must be made in terms of clearance and Vd (5) . Most often, it is a difference in clearance which itself can be explained in terms of fu (degree of plasma protein binding) and intrinsic clearance, etc. Moreover, authors should be aware that the literature is full of miscalculations. For example for reference 23 relating to the Red-Tailed Hawks  and Great Horned Owls , it is reported in table 3   that half- lives for Red-Tailed Hawks are  19.4, 11 and 8.9 h for the IV, IM and oral routes of administration  respectively. The Authors of this review should have immediately noted that the value reported by the IM route (and also orally) are false because conceptually, a half-life time cannot be shorter by the IM route than by the IV route! As such, comparisons of the half-lives by IM route of Red-Tailed Hawks with Great Hornet Owl doesn't make sense. HARRENSTIEN's article ( ref 23) is the typical example of an article published in a journal with no expertise in PK and which requires critical analysis to be a valid source of primary data for the present secondary  scoping review . For example, the fact that Harrenstein et al confuse smoothing and fitting (see their plots), do not report the value of the clearance which is the main PK parameter, do not specify the nature of the volume of distribution that they calculated… should have attracted the attention of the Authors of the present review to the poor quality of this article. To reports results of HARRENSTIEN'  et al without putting them into perspective adds confusion as is the case for lines 153-155 which affirm " The half -life of enrofloxacin in the red-tailed hawk varied considerably between the intramuscular (IM), intravenous and oral (PO) administration routes. ] The reason for this is unclear ,… Authors  should have simply said that the calculation of the half-life was faulty and that it should not be taken into account (or more simply, Authors could have  ignored this wrong figures)

Line 247 it is reported “ Marbofloxacin seems to follow non-linear kinetics as drug half-life varies within the same species (Eurasian buzzard, Japanese quail, common pheasant and blue-and-gold macaw) according to the dose and route of administration used ”that  is misleading. Table 5 does not report such large differences. In addition, there are many other reasons for having differences in half-life values, including different  LOQs for the  analytical techniques, the duration of sampling, more or less robust or faulty calculation methods ... I recommend reading reviews which explains what is a half-life (5).  

 I stop here  my criticisms there but there would be others to make but the authors must first completely reconsider their article so that a referee can improve it usefully

References

  1. Vermeulen B, De Backer P, Remon JP. 2002. Drug administration to poultry. Adv Drug Delivery Rev 54:795–803.
  2. Toutain PL, Ferran A, Bousquet-Mélou A. 2010. Species Differences in Pharmacokinetics and Pharmacodynamics, p. 19–48. In Cunningham, F, Elliott, J, Lees, P (eds.), Comparative and Veterinary Pharmacology. Springer Berlin Heidelberg, Berlin, Heidelberg.
  3. Toutain P, Pelligand L, Lees P, Bousquet - Mélou A, Ferran AA, Turnidge JD. 2021. The pharmacokinetic/pharmacodynamic paradigm for antimicrobial drugs in veterinary medicine: Recent advances and critical appraisal. Jvet Pharmacol Therap 44:172–200.

Author Response

Reviewer 3

The title of this review is attractive but its content is rather disappointing. It is rather a series of tables compiling figures reported in the literature without any real critical analysis of what has been published. A critical review in this field requires a high level of expertise in pharmacokinetics (PK) and also in pharmacodynamics. This is all the more important since the primary literature on the PK of exotic birds is often generated by non-specialists and the risk is to peddle in this review results that are more or less flawed or poorly interpreted (see my specific comments).

Moreover, I believe that as a preliminary, the authors should recall some basic notions on the physiological, metabolic and behavioural specificities of exotic birds having possible consequences on drugs’ PK. They should explain whether the particularities reported in poultry, in particular on the absorption and elimination of drugs (1), (2) are also found or not in exotic birds. For example, does the existence of a reptilian-type kidney with the possibility or not associated to a first-pass effect for drugs administered intramuscularly in the thigh? if yes, this require to qualify in this review the site of IM administration because bioavailability could be very different depending of the IM site of administration.

We thank the reviewer for this feedback. We have now added Section 3 to provide more information on this.

Similarly, behavioral particularities such as the fact that birds (atleast poultry) only drink during the lighting period and that water consumption is dependent of the ambient temperature should be reported and discussed. This is because it is a problem for the

oral route of administration when one wants to treat an infection with a time-dependent antibiotic. They could learn from Vermelen 's excellent article (1). It is also from the experience of this referee that certain PK concepts are not well understood by non-specialist and the Authors must explain them.

In-water and in-food administration of medication is not a focus of this review as these routes are not common in non-poultry avian species. As direct routes of dosing (PO, SC, IM) are more important in non-poultry avian species, behavioral particularities are less relevant here.

Line 41-47: the Authors recommend making Antimicrobial Susceptibility Testing (AST) which here is not welcome. Indeed the Clinical Breakpoints (CBP) which are defined by major international organizations such as EUCAST, CLSI, USCAST, etc. are defined for each species and for well-defined dosages. However, these dosages are unknown for exotic birds! As such, the interpretation of an AST by taking CBP from another species can be very misleading. Rather, the authors could give for the main pathogens of interest in exotic birds the corresponding ECOFFs ( i.e. the epidemiological cut-off) which are specific to a given bacterial species and which have the same value in all animal species (see these ECOFFs on the EUCAST site: https://www.eucast.org/mic_distributions_and_ecoffs/). And it is the ECOFFs that can be used to interpret an AST in exotic birds in the absence of CBP. This is what is recommended by VetCAST, the veterinary subgroup of EUCAST. By following this approach, the authors could compare the plasma concentrations obtained with these ECOFFs (MIC of wild population) and thus make a first judgment on the validity of the dosages currently recommended for exotic birds (see the EUCAST/VetCAST website and the associated publications https://www.eucast.org /ast_of_veterinary_pathogens/ ).

With the advancement in the field of antibiotics, we acknowledge and understand the limitations of AST. A systematic review-like approach was adopted in searching for the relevant articles based on the search concept of the specific antibiotic name and avian species. 90% of the included studies were PK studies performed using healthy animals. The concentration achieved was used to predict efficacy based on whether target MIC would have been achieved or not. This was a relevant measure of efficacy at the time of practice and publication of these studies. The MIC values were typically obtained from other studies by the authors and if the antibiotic targets different microorganism, the highest MIC would be chosen as a surrogate on whether efficacy would be achieved with the concentrations observed. However, to acknowledge this deficiency and increase awareness of the importance using clinical breakpoints including ECOFFs in future antibiotic studies, a paragraph has been added in the introduction:

“Moreover, the use of clinical breakpoints is the new recommended standard for guiding antibiotic use. Clinical breakpoints should be based on an epidemiological cut-off value (ECOFF), a PK/PD breakpoint obtained from pre-clinical pharmacokinetic and wherever possible, a clinical cut-off. However, a paucity of data in veterinary medicine, especially avian veterinary medicine, precludes widespread use of this modality.”

We have also added Section 5 to provide this background to orientate readers on why MIC values were reported.

Another important point that is not addressed in this paper is that all PK/PD indices must be calculated in terms of free, not total plasma concentrations (3) . I am very concerned that this information on plasma protein binding of antimicrobials is not available and the Authors must discuss it, especially in their last part entitled “ recommendation for research ”.

Line 45, 74 ...: PK/PD index are not to compute efficacy but only to predict efficacy.

Furthermore, the Cmax/MIC index is no longer used either by EUCAST or by USCAST. According to USCAST, “Given the transient nature of Cmax and the ability to estimate AUC with greater precision than Cmax, AUC:MIC rather than Cmax:MIC ratio targets for efficacy were characterized and considered for PK-PD target attainment analyses, see https:/ /app.box.com/s/1hxc8inf8u3rranwmk3efx48upvwt0ww )

It is beyond the scope of this review to calculate parameters not reported in the original manuscripts.

Line 85: it is true that the optimal target is unknown but a target of 13 h is very unlikely. Indeed , it is poorly understood that a target of 24 h just means that the average free plasma concentration over 24h and in steady-state conditions, is equal to the MIC and a
target of 13 h mean that efficacy would be achieved with an average plasma concentration equal to only half the MIC!

Line 104-108: The Authors reported different PK parameters and variables without ranking them. Actually, the most important of the PK parameters is the plasmatic clearance whereas the AUC is only a hybrid variable depending on dose, clearance and bioavailability for the extravascular routes of administration. Authors needs to precisely qualify reported PK parameters and their interpretation. Just taking the volume of distribution (Vd) as an example, the Authors claim to report “the” volume of distribution when there are actually three volumes of distribution that are routinely reported in the literature (Vc, Vss and Varea) with definitions and interpretations that are different (4) moreover, for the extravascular routes, what is calculated is a V/F, i.e. an apparent volume of distribution which depends on the bioavailability (symbol F, not BA). This V/F is therefore not a parameter but a contextual variable with no interest in being reported in this review. Let's just take as an example the volume of distribution of enrofloxacin in Houbara bustard. Reported values for the IV, IM and oral routes are 2.98, 3.18 and 5.12 L/Kg. In fact, the only relevant value (because it corresponds to a primary parameter) is that of the IV route, the other values being variables, not parameters which depend on F. this explains the greater value of Vd for the oral route because its bioavailability is the lowest.

The purpose of Section 3 was to describe the search methods and data extracted. Various PK parameters were extracted from the literature where available.

Line 132: I see no evidence of difference in the half-life times of enrofloxacin in common ostrich in table 3.

This has been removed.

More importantly, the whole discussion on the interpretation of differences in half-life needs to be reconsidered. Authors should realize that the half-life
is a hybrid parameter that depends on clearance and volume of distribution (here varea ) and that the first level of interpretation to explain observed differences must be made in terms of clearance and Vd (5) . Most often, it is a difference in clearance which itself
can be explained in terms of fu (degree of plasma protein binding) and intrinsic clearance, etc. Moreover, authors should be aware that the literature is full of miscalculations. For example for reference 23 relating to the Red-Tailed Hawks and Great Horned Owls , it is reported in table 3 that half- lives for Red-Tailed Hawks are 19.4, 11 and 8.9 h for the IV, IM and oral routes of administration respectively. The Authors of this review should
have immediately noted that the value reported by the IM route (and also orally) are false because conceptually, a half-life time cannot be shorter by the IM route than by the IV route! As such, comparisons of the half-lives by IM route of Red-Tailed Hawks with
Great Hornet Owl doesn't make sense. HARRENSTIEN's article (ref 23) is the typical example of an article published in a journal  with no expertise in PK and which requires critical analysis to be a valid source of primary data for the present secondary scoping review . For example, the fact that Harrenstein et al confuse smoothing and fitting (see their plots), do not report the value of the clearance which is the main PK parameter, do not specify the nature of the volume of distribution that they calculated... should have attracted the attention of the Authors of the present review to the poor quality of this article. To reports results of HARRENSTIEN' et al without putting them into perspective adds confusion as is the case for lines 153-155 which affirm " The half -life of enrofloxacin in the red-tailed hawk varied considerably between the intramuscular (IM), intravenous and oral (PO) administration routes. ] The reason for this is unclear ,... Authors should have simply said that the calculation of the half-life was faulty and that it should not be taken into account (or more simply, Authors could have ignored this wrong figures)

We thank the reviewer for this feedback. We have included a brief introduction of the various PK parameters to provide some background for the readers before delving into data interpretation. While various PK parameters were extracted and presented in the Tables in the manuscript, due to the heterogeneity of methods in deriving these parameters and presentation of results by the various studies as pointed out by the reviewer, we have opted to focus the discussion for PK trends on the half-life of drugs, since it is a composite of CL and V. Any differences in drug half-life could be due to differences in CL or V which could be due to differences in clearance processes, or even impacted by differences in volume of distribution due to interspecies differences in fat or water composition of the animal and its interaction with the lipophilicity/hydrophilicity of the drug. We agree with the reviewer that the quality of the studies may be poor and hence interpretation of data should be done with caution. We have since removed these discussion points that may not be very meaningful.

Line 247 it is reported “ Marbofloxacin seems to follow non-linear kinetics as drug half-life varies within the same species (Eurasian buzzard, Japanese quail, common pheasant and blue-and-gold macaw) according to the dose and route of administration used ”that is misleading. Table 5 does not report such large differences. In addition, there are many other reasons for having differences in half-life values, including different LOQs for the analytical techniques, the duration of sampling, more or less robust or faulty calculation methods ... I recommend reading reviews which explains what is a half-life (5). I stop here my criticisms there but there would be others to make but the authors must first completely reconsider their article so that a referee can improve it usefully

We thank the reviewer for this feedback. This discussion has been revised.

Round 2

Reviewer 1 Report

Thank you.

Author Response

Thank you very much.

Reviewer 2 Report

Dear Authors,

Thank you for your responses and efforts. 

Best of luck!

Author Response

Thank you very much.

Reviewer 3 Report

The revised version has been greatly improved and no longer contains any major errors. the only point that I had raised and which is not presented and discussed in this version is that of plasma protein binding. The authors must state that only the free antibiotic concentrations are effective and that the plasma concentrations reported in the literature are all total concentrations. They must suggest that the measurement of plasma protein binding which may show significant interspecific variabilities is a research priority for this community.

Author Response

The requested information has been added in line 128-131:

"Lastly, most current literature only report total concentrations instead of plasma concentrations. Future studies should include the measurement of plasma protein binding which may show significant interspecific variabilities as only the free antibiotic concentrations are clinically relevant."